# Adaptive Cartesian Meshes for Atmospheric Single-Column Models, a study using Basilisk 18-02-16

J.Antoon van Hooft[1], Stéphane Popinet[2], and Bas J.H. van de Wiel[1]

[1]Delft University of Technology, Department of Geoscience and Remote Sensing, Delft, The Netherlands
[2]Sorbonne Université, Centre National de Recherche Scientifique, UMR 7190, Institut Jean Le Rond D'Alembert, F-75005, Paris, France

**Correspondence:** Antoon van Hooft (j.a.vanhooft@tudelft.nl)

**Abstract.** It is well known that the representation of certain atmospheric conditions in climate and weather models can still suffer from the limited grid resolution that is facilitated by modern-day computer systems. Herein we study a simple one-dimensional analogy to those models by using a Single-Column Model description of the atmosphere. The model employs an adaptive Cartesian mesh that applies a high-resolution mesh only when and where it is required. The so-called adaptive-grid model is described and we report on our findings obtained for tests to evaluate the representation of the atmospheric boundary layer, based on the first two GABLS intercomparison cases. The analysis shows that the adaptive-grid algorithm is indeed able to dynamically coarsen and refine the numerical grid whilst maintaining an accurate solution. This is an interesting result as in reality, transitional dynamics (e.g. due to the diurnal cycle or due to changing synoptic conditions) are the rule rather than the exception.

## 1 Introduction

Single-Column Models (SCMs) are often used as the building blocks for Global (or General) Circulation Models (GCMs). As such, many of the lessons learned from SCM development can be inherited by GCMs and hence the evaluations of SCMs receive considerable attention by the geoscientific model development community (see e.g. Neggers et al., 2012; Bosveld et al., 2014; Baas et al., 2017). In this work, we present a SCM that employs an adaptive Cartesian mesh that can drastically reduce the computational costs of such models, especially when pushing the model's resolution. The philosophy is inspired by recently obtained results on the evolution of atmospheric turbulence in a daytime boundary layer using three-dimensional (3D) adaptive grids. As promising results were obtained for turbulence-resolving techniques such as Direct Numerical Simulations and Large-eddy Simulation (LES), herein we explore whether similar advancements can be made with more practically oriented techniques for the numerical modelling of the atmosphere. As such, the present model uses Reynolds-averaged Navier-Stokes (RANS) techniques to parameterize the vertical mixing processes due to turbulence (Reynolds, 1895), as is typical in weather and climate models.

The discussion of limited grid resolution is present in many studies of SCMs and GCMs. A prominent example is the nocturnal cumulus-cloud case (Wyant et al., 2007): whereas a high resolution mesh is required for capturing the processes at the cloud interface, a coarser resolution may be used for the time when the sun has risen and the cloud has been dissolved. More generally speaking, virtually all of the atmospheric dynamics that require a relatively high-resolution grid for their representation in numerical models are localized in both space and in time. The issue is made more difficult to tackle by the fact that their spatio-temporal localization is typically not known a priori (e.g. the height and strength of a future inversion layer). Therefore, the pre-tuned and static-type grids that most operational GCMs use (virtually all) are not flexible enough to capture all dynamical regimes accurately or efficiently. This also puts a large strain on the used closures for the sub-grid scale processes. In order to mitigate this challenge, GCMs that employ a so-called adaptive grid have been explored in the literature. Here the grid resolution adaptively varies in both space and time, focussing the computational resources to where and when they are most necessary. Most notably, the innovative works of Jablonowski (2004), Jablonowski et al. (2009) and St-Cyr et al. (2008) report on the usage of both Cartesian and non-conforming three-dimensional adaptive grids and clearly demonstrate the potential of grid adaptivity for GCMs. Inspired by their works, we follow a 1D SCM approach and aim to add to their findings, using different grid-adaptative formulations and solver strategies. Since SCMs do not resolve large-scale atmospheric circulations, the analysis herein focusses on the representation of the Atmospheric Boundary Layer (ABL).

Over the years, the computational resources that are available to run computer models have increased considerably (Schaller, 1997). This has facilitated GCMs to increase their models' spatial resolution, enabling to resolve the most demanding processes with increased grid resolutions. However, it is important to realize that the (spatial and temporal) fraction of the domain that benefits most from an increasing maximum resolution necessarily decreases as separation of the modelled spatial scales increases (Popinet, 2011). This is because the physical processes that warrant a higher-resolution mesh are virtually never space filling. E.g. the formation phase of tropical cyclones is localized in both space and time and is characterized by internal dynamics that evolve during the formation process. By definition, with an increasing scale separation, only an adaptive-grid approach is able to reflect the effective (or so-called fractal) dimension of the physical system in the scaling of the computational costs (Popinet, 2011; Van Hooft et al., 2018). This is an aspect where the present adaptive-grid approach differs from for example, a dynamic-grid approach (Dunbar et al., 2008), that employs a fixed number of grid cells that needs to be predefined by the user. This work employs a similar method for grid adaptation as presented in the work of Van Hooft et al. (2018) on 3D-turbulence-resolving simulations of the ABL. As such, this work is also based on the adaptive-grid toolbox and built-in solvers provided by the 'Basilisk' code (http://basilisk.fr).

We test our model with the well established cases defined for the first two GABLS intercomparison projects for SCMs. As part of the Global Energy and Water cycle EXchanges (GEWEX) modelling and prediction panel, the GEWEX ABL Study (GABLS) was initiated in 2001 to improve understanding of the atmospheric boundary layer processes and their representation in models. Based on observations during field campaigns, a variety of model cases has been designed and studied using both LES and SCMs with a large set of models using traditional static-grid structures. An overview of the results and their interpretation for the first three intercomparison cases are presented in the work of Holtslag et al. (2013). Here we will test the present adaptive-grid SCM based on the first two intercomparison cases, referred to as GABLS1 and GABLS2. These cases

were designed to study the model representation of the stable boundary layer and the diurnal cycle, respectively. Their scenarios prescribe idealized atmospheric conditions and lack the complete set of physical processes and interactions encountered in reality. At this stage within our research, the authors consider this aspect to be an advantage, as the present SCM model does not have a complete set of parameterizations for all processes that are typically found in the operational models (see e.g. Slingo, 1987; Grell et al., 2005)).

This paper is organized as follows, the present SCM is discussed in more detail in Sect. 2. Based on the results from a simplified flow problem, Sect. 3 starts with an analysis of the used numerical methods and the grid adaptation strategy. Model results for ABL-focussed cases that are based on the first two GABLS intercomparison scenarios are also presented in Sect. 3. Finally, a discussion and conclusions are presented in Sect. 4.

## 2   Model Overview

As we focus on the merits of grid adaptivity in this study on SCMs and not on the state-of-the-art closures for the vertical transport phenomena, we have opted to employ simple and well-known descriptions for the turbulent transport processes. More specifically, the present model uses a stability-dependent, first-order, local, $K$-diffusivity closure as presented in the work of Louis et al. (1982) and Holtslag and Boville (1993). For the surface-flux parameterizations we again follow the formulations in the work of Holtslag and Boville (1993). However, to improve the representation of mixing under stable conditions, an alteration is made to the formulation of the so-called stability-correction function under stably-stratified conditions. Based on the work of England and McNider (1995), we use a so-called short-tail mixing function. The used closures for the turbulent transport are summarized next. The upward *surface* fluxes ($F$) of the horizontal velocity components ($u, v$), the potential temperature ($\theta$) and specific humidity $q$ are evaluated as:

$$F_u = -C_M U_1 u_1, \tag{1a}$$

$$F_v = -C_M U_1 v_1, \tag{1b}$$

$$F_\theta = -C_H U_1 \left( \theta_1 - \theta_0 \right), \tag{1c}$$

$$F_q = -C_H U_1 \left( q_1 - q_0 \right), \tag{1d}$$

Where $U$ is the wind-speed magnitude and indices $0$ and $1$ refer the to values at the surface and the first model level, respectively. The surface transport coefficients are,

$$C_M = C_N f_{s,M}(\mathrm{Ri_b}), \tag{2a}$$

$$C_H = C_N f_{s,H}(\mathrm{Ri_b}), \tag{2b}$$

with $\mathrm{Ri_b}$ the surface bulk Richardson number, that is defined as,

$$\mathrm{Ri_b} = \frac{g}{\theta_{v,\mathrm{ref}}} \frac{z_1 \left(\theta_{v,1} - \theta_{v,0}\right)}{U_1^2}, \tag{3}$$

where $g$ is the acceleration due to gravity, $\theta_v$ is the virtual potential temperature and $\theta_{v,\mathrm{ref}}$ is a reference temperature whose value is taken as a scenario-specific constant. Equation 3 assumes that $\theta_v$ is related to the buoyancy ($b$) (Boussinesq, 1897) via $g$ and $\theta_{v,ref}$ according to $b = g/\theta_{v,ref}(\theta_v - \theta_{v,ref})$ . The virtual potential temperature is related to the potential temperature ($\theta$) and specific humidity ($q$) according to,

$$\theta_v = \theta \left(1 - \left(1 - \frac{R_v}{R_d}\right) q\right), \tag{4}$$

with $R_v/R_d = 1.61$ the ratio of the gas constants for water vapour and dry air (Emauel, 1994; Heus et al., 2010). The so-called neutral exchange coefficient ($C_N$) is calculated using,

$$C_N = \frac{k^2}{\ln\left((z_1 + z_{0,M})/z_{0,M}\right)^2}, \tag{5}$$

with $k = 0.4$ the Von Karman constant, $z_1$ the height of lowest model level and $z_{0,M}$ is the roughness length for momentum. For the cases studied in this work, the roughness length for heat is assumed to be identical to $z_{0,M}$. The stability correction functions for the surface transport of momentum and heat ($f_{s,M}, f_{s,H}$) are,

$$f_{s,M}(\mathrm{Ri_b}) = \begin{cases} 0, & \mathrm{Ri_b} \geq 0.2, \\ \left(1 - \frac{\mathrm{Ri_b}}{0.2}\right)^2, & 0 \leq \mathrm{Ri_b} < 0.2, \\ 1 - \frac{10\mathrm{Ri_b}}{1 + 75 C_N \sqrt{((z_1 + z_{0,M})/z_{0,M})\|\mathrm{Ri_b}\|}}, & \mathrm{Ri_b} < 0, \end{cases} \tag{6a}$$

$$f_{s,H}(\mathrm{Ri_b}) = \begin{cases} f_{s,M}(\mathrm{Ri_b}), & \mathrm{Ri_b} \geq 0, \\ 1 - \frac{15\mathrm{Ri_b}}{1 + 75 C_N \sqrt{((z_1 + z_{0,M})/z_{0,M})\|\mathrm{Ri_b}\|}}, & \mathrm{Ri_b} < 0, \end{cases} \tag{6b}$$

which conclude the description of the surface fluxes. The vertical flux ($\overline{w'a'}$) of a dummy variable $a$ due to turbulence within the boundary layer is based on a local diffusion scheme and is expressed as,

$$\overline{w'a'} = -K \frac{\partial a}{\partial z}, \tag{7}$$

where $K$ is the so-called eddy diffusivity,

$$K = l^2 S f(\mathrm{Ri}). \tag{8}$$

$l$ represents an effective mixing length,

$$l = \min\left(kz, l_{bl}\right),\tag{9}$$

with $l_{bl}$ is the Blackadar length scale, we use, $l_{bl} = 70m$ (Holtslag and Boville, 1993). $S$ is the local wind-shear magnitude,

$$S = \sqrt{\left(\frac{\partial u}{\partial z}\right)^2 + \left(\frac{\partial v}{\partial z}\right)^2}\tag{10}$$

and $f(\mathrm{Ri})$ is the stability correction function for the vertical flux,

$$f(\mathrm{Ri}) = \begin{cases} 0, & \mathrm{Ri} \geq 0.2, \\ \left(1 - \frac{\mathrm{Ri}}{0.2}\right)^2, & 0 \leq \mathrm{Ri} < 0.2, \\ \sqrt{1 - 18\mathrm{Ri}}, & \mathrm{Ri} < 0, \end{cases}\tag{11}$$

i.e. based on the gradient Richardson number,

$$\mathrm{Ri} = \frac{g}{\theta_{v,ref}} \frac{\partial \theta_v / \partial z}{S^2}.\tag{12}$$

The authors of this work realize that there have been considerable advancements on the representation of mixing under

unstable conditions in the past decades, e.g non-local mixing (Holtslag and Boville, 1993) and turbulent-kinetic-energy-based closures (see e.g., Mellor and Yamada, 1982; Lenderink and Holtslag, 2004). Therefore, we would like to note that such schemes are compatible with the adaptive-grid approach and they could be readily employed to improve the physical descriptions in the present model. From an implementations' perspective, those schemes would not require any grid-adaptation specific considerations when using the Basilisk code.

For time integration; we recognize a reaction-diffusion-type equation describing the evolution of the horizontal wind components and scalar fields such as the virtual potential temperature and specific humidity ($q$). For a variable field $s(z,t)$, we write,

$$\frac{\partial s}{\partial t} = \frac{\partial}{\partial z}(K\frac{\partial}{\partial z}s) + r.\tag{13}$$

Where $r$ is a source term and $K$ is the diffusion coefficient c.f. Eq. 8. Using a mixed implicit-explicit first-order-accurate time

discretization for the diffusive term and an explicit time integration for the source term ($r$), with time step $\Delta t$ separating the solution $s^n$ and $s^{n+1}$, this can be written,

$$\frac{s^{n+1} - s^n}{\Delta t} = \frac{\partial}{\partial z}(K^n \frac{\partial}{\partial z}s^{n+1}) + r^n.\tag{14}$$

Rearranging the terms we write,

$$\frac{\partial}{\partial z}(K^n \frac{\partial}{\partial z}s^{n+1}) - \frac{s^{n+1}}{\Delta t} = -\frac{s^n}{\Delta t} - r^n,\tag{15}$$

to obtain a Poisson-Helmholtz equation for $s^{n+1}$, using the eddy diffusivity calculated from the solution $s^n$ ($K^n$). Eq. 15 is solved using a multigrid strategy, employing a finite-volume-type second-order-accurate spatial discretization (Popinet, 2017a, b). The source term $r$ in Eq. 13 is defined using different formulations for the various scalar fields in our model. For $\theta$ and $q$, the source term $r$ concerns the tendency in the lowest grid level due to the surface fluxes ($F$, see Eqs. 1, $r_{F_s}$) and the effect of large scale synoptic divergence ($r_w$) according to the vertical velocity $w$ (i.e. prescribed for the GABLS2 case). We write for a dummy variable $s$,

$$r_{w,s} = -w\frac{\partial s}{\partial z} \tag{16}$$

For the horizontal velocity components $(u,v)$ the corresponding source terms (i.e. $r_{F_s}$ and $r_w$) are also taken into account and supplemented with the additional source term $r_{\boldsymbol{\nabla}_h P,f}$, that concerns the horizontal pressure-gradient-forcing vector (i.e. $-\frac{1}{\rho}\boldsymbol{\nabla}_h P$, for air with a density $\rho$) and the Coriolis-force term according to the local Coriolis parameter $f$. For the horizontal velocity vector $\boldsymbol{u} = \{u,v,0\}$ we write,

$$r_{\boldsymbol{\nabla}_h P,f} = \frac{-\boldsymbol{\nabla}_h P}{\rho} - f\left(\hat{\boldsymbol{k}} \times \boldsymbol{u}\right), \tag{17}$$

where '$\times$' represents the cross product operator and $\hat{\boldsymbol{k}} = \{0,0,1\}$ the unit vector in the vertical direction. In this work we adopt the commonly used strategy to introduce a velocity vector that known as the geostrophic wind ($\boldsymbol{U_{geo}}$), according to,

$$\boldsymbol{U_{geo}} = \frac{\hat{\boldsymbol{k}}}{\rho f} \times \boldsymbol{\nabla}_h P. \tag{18}$$

The most prominent feature of the SCM presented in this work is that it adaptively coarsens and refines the grid resolution based on the evolution of the solution itself. As mentioned in the introduction, the associated grid-adaptation algorithm is the same as described in Van Hooft et al. (2018). Here we only briefly discuss the general concept.

Apart from the imperfect representation of the physical aspects of a system in numerical models, additional errors naturally arise due to the spatial and temporal discretization. In general, a finer resolution corresponds to a more accurate solution and a simulation result is considered to be 'converged' when the numerically obtained solution and the statistics of interest do not crucially depend on the chosen resolution. The aim of the grid-adaptation algorithm is to dynamically coarsen and refine the mesh so that the errors due to the spatial discretization remain within limited bounds and to be *uniformly distributed* in both space and time. For our adaptive approach this requires, (1) an algorithm that evaluates a local estimate of the discretization error in the representation of selected solution fields ($\chi_a$ for a field '$a$') and (2), a corresponding error threshold ($\zeta_a$) that determines if a grid cell's resolution is either too coarse (i.e. $\chi_a > \zeta_a$), too fine (i.e. $\chi_a < 2\zeta_a/3$), or just fine. Grid adaptation can then be carried out accordingly and the solution values of new grid cells can be found using interpolation techniques. A cell is refined when the estimated error for at least one selected solution field exceeds it's refinement criterion and a cell is coarsened when it is considered to be 'too fine' for all selected solution fields. The 'error estimator' ($\chi$) is based on a so-called multi-resolution analysis that is formally linked to wavelet thresholding. The algorithm aims to estimate the magnitude of higher-order contributions in the spatial variability of the solution that are not captured by the solver's numerical schemes.

Consistent with the second-order spatial accuracy of the solver's numerical schemes (Popinet, 2017b), the algorithm employs a second-order accurate wavelet-based error estimate. In practice, grid refinement will typically occur at the locations where the solution is highly 'curved', whereas those areas where the solution fields vary more 'linearly' in space are prone to coarsening. The error threshold, or so-called refinement criterion $\zeta$, is defined by the user. Noting that similar to the pre-tuning of the fixed-in-time grids as is common in most SCMs, the balance between accuracy and the required computational effort remains at the discretion of the model's user.

For the cases in this work that focus on the ABL (i.e. in Sect 3.2 and 3.3), the dynamics are governed by the wind ($\boldsymbol{U} = (u, v)$) and the virtual potential temperature ($\theta_v$), hence we base the refinement and coarsening of the grid on a second-order-accurate estimated error associated with the representation of these discretized fields. Based on trial and error, we set the corresponding refinement thresholds,

$$\zeta_{u,v} = 0.25 \text{ m/s}, \tag{19}$$

$$\zeta_{\theta_v} = 0.5 \text{ K}, \tag{20}$$

for both of the horizontal wind components and virtual potential temperature, respectively. These values are the result of a choice by the authors that aims to strike an arbitrary balance between the accuracy of the solution and the computational effort required to run the model. Note that a similar (arbitrary) balance needs also to be found when static grids are employed. For a simple flow set-up, Sect. 3.1 presents an example convergence study to show the effects of using different refinement criteria on the accuracy of the obtained solutions.

Grid adaptation is carried out each time step. The tree-based anisotropic-grid structure in Basilisk facilitates a convenient basis for the multi-resolution analysis and the subsequent refinement and coarsening of cells at integer levels of refinement. This entails that the spatial resolution can vary by factors of two (Popinet, 2011). For the adaptive-grid runs presented in this paper, the time spent in the actual grid assessment and adaptation routines is less than than 5% of the total wall-clock time (see table 1).

Apart from the Ekman-spiral case in Sect. 3.1, the physical time step in the ABL-focussed cases is adaptively varied between $2$ sec. and $15$ sec. based on the convergence properties of the aforementioned iterative solver. Noting that these values are rather small compared to existing GCMs that often employ higher-order-accurate time-integration schemes. Additionally, the correlation of spatial and temporal scales warrants a smaller time step, since the present model employs a higher maximum vertical resolution compared to that of an operational GCM. The solver's second-order spatial accuracy is validated and the performance is accessed for a simple flow set-up in Sect. 3.1. For the exact details of the model set-ups for the cases presented in this paper, the reader is referred to the case-definition files (in legible formatting). Links are provided to their online locations in table 1.

## 3  Results

### 3.1   The Laminar Ekman spiral and grid adaptation

Before we focus our attention on cases that concern the ABL, this section discusses the philosophy of the used grid adaptation strategy based on the analysis of a one-dimensional (1D) *laminar* Ekman-flow set-up. This simple and clean set-up enables to quantify numerical errors explicitly and test the solver's numerical schemes. The aim of this section is to show that the grid-adaptation strategy and the accompanying refinement criteria provide a consistent and powerful framework for adaptive mesh-element-size selection. Results are presented for both an equidistant-grid and the adaptive-grid approach. The case describes a neutrally-stratified fluid with a constant diffusivity for momentum ($K$) given by the kinematic viscosity $\nu$ and density $\rho$ in a rotating frame of reference with respect to the Coriolis parameter $f$. A flow is forced by a horizontal pressure gradient $(-\boldsymbol{\nabla_h} P)$ according to Eq. 18 using $\boldsymbol{U_{geo}} = \{U_{geo}, 0\}$, over a no-slip bottom boundary (located at $z_{bottom} = 0$). Assuming that the velocity components converge towards the geostrophic wind vector for $z \to \infty$ and vanish at the bottom boundary, there exists an analytical, 1D, steady solution for the horizontal wind component profiles $(u_E(z), v_E(z))$, that is known as the Ekman spiral;

$$u_E = U_{geo}\left(1 - e^{-\gamma z}\cos(\gamma z)\right), \tag{21}$$

$$v_E = U_{geo} e^{-\gamma z}\sin(\gamma z), \tag{22}$$

with $\gamma$ the so-called inverse Ekman depth, $\gamma = \sqrt{f/(2\nu)}$. We choose numerical values for $U_{geo}, \gamma$ and $f$ of unity in our set-up and present the results in a dimensionless framework. The solution is initialized according to the exact solution and we set boundary conditions based on Eqs. 21 and 22. Equation 13 is solved numerically for both $u$ and $v$ components, on a domain with height $z_{top} = 100\gamma^{-1}$. The simulation is run until $t_{end} = 10f^{-1}$, using a fixed time step $\Delta t = 0.01f^{-1}$. The time step is chosen sufficiently small such that the numerical errors are dominated by the spatial discretization rather than by the time-integration scheme. During the simulation run, discretization errors alter the numerical solution from it's exact, and analytically steady, initialization. For all runs, the diagnosed statistics regarding the numerical solutions that are presented in this section have become steady at $t = t_{end}$.

The spatial-convergence properties for the equidistant-grid solver are studied by iteratively decreasing the used (equidistant) mesh-element sizes ($\Delta$) by factors of two and we monitor the increasing fidelity of the solution at $t = t_{end}$ between the runs. Therefore, based on the analytical solution, a local error ($\epsilon_{u,v}$) of the numerically obtained solution ($u^n, v^n$) within each grid cell is diagnosed and is defined here as:

$$\epsilon_a = \|a^n - \langle a_E \rangle\|, \tag{23}$$

where $a$ is a dummy variable for $u$ and $v$, $\langle a_E \rangle$ is the grid-cell-averaged value of the analytical solution ($a_E$) and $a^n$ the value of the numerical solution within the cell. Noting that $a^n$ also represents the grid-cell-averaged value in our finite-volume approach. Figure 1a shows the results for all runs and compares the used grid resolution ($\Delta$) with the error $\epsilon_{u,v}$. It appears that the observed range of $\epsilon$-values is large and typically spans 10 orders of magnitude, with a lower bound defined by the 'machine

precision' (i.e. $\approx 10^{-15}$ for double-precision floating-point numbers). This wide range can be explained by the fact that the Ekman spiral is characterized by exponentially decreasing variation with height (see Eqs. 21, 22) and hence the equidistant grid may be considered overly refined at large $z$. This illustrates that, for a given solver formulation, the error in the solution is not directly dictated by the mesh-element size, but also depends on *the local shape of the numerical solution itself*. This poses a challenge for the pre-tuning of meshes applied to GCMs, where a balance need to be found between accuracy and computational speed performance. The solution of a future model run is not known beforehand and hence the tuning of the grid typically relies heavily on experience, empiricism and a-priori knowledge. This motivates to apply the method of error *estimation* in the representation of a discretized solution field as described in Popinet (2011) and Van Hooft et al. (2018). For both velocity components, this estimated error ($\chi_{u,v}$) is evaluated at the end of each simulation run for each grid cell and is plotted against the corresponding actual error ($\epsilon_{u,v}$) in Fig. 1b. It seems that for this virtually steady case, there is a clear correlation between the diagnosed (instantaneous) $\chi$-values and the $\epsilon$-values that have accumulated over the simulation run time. Even though the correlation is not perfect, it provides a convenient and consistent framework for a grid adaptation algorithm. As such, a second convergence test for this case is performed using a variable-resolution grid within the domain. The mesh is based on the aforementioned adaptive-grid approach. For these runs, we iteratively decrease the so-called refinement criterion ($\zeta_{u,v}$) by factors of two between the runs and monitor the increasing fidelity of the numerically obtained solution for all runs. The refinement criterion presets a threshold value ($\zeta$) for the estimated error $\chi$ that defines when a cell should be refined ($\chi > \zeta$) or alternatively, when it may be coarsened ($\chi < 2\zeta/3$). Figure 2a presents the results and compares the used local grid resolution against $\epsilon_{u,v}$ for the various (colour-coded) runs. It appears that for all separate runs, the algorithm employed a variable resolution mesh and that this has resulted in a smaller range of the local error in the solution ($\epsilon$), as compared to the equidistant-grid cases. The local error in the solution is also compared against the wavelet-based estimated error in the representation of the solution fields in Fig. 2b. Compared to the results from the equidistant-grid approach as presented in Fig. 1b), the spread of the $\chi$ and $\epsilon$ values is relatively small for the separate runs when the adaptive-grid approach is used. The most prominent reason for the finite spread is that the error ($\epsilon$) was diagnosed after 1000 time steps. This facilitated errors in the solution that arise in the solution at a specific location (with a large $\chi$-value) to 'diffuse' over time towards regions where the solution remains to be characterized by a small $\chi$-value (not shown). Also, since $u$ and $v$ are coupled (due to the background rotation), local errors that arise in the solution for $u$ 'pollute' the $v$-component solution, and vice versa. Furthermore, a spread is expected because the tree-grid structure only allows the resolution to vary by factors of two (Popinet, 2011).

Finally, the global convergence characteristics and the speed performance of the two approaches are studied. The global error ($\eta$) in the numerically obtained solution is evaluated as,

$$\eta = \int_{z_{bottom}}^{z_{top}} (\epsilon_u + \epsilon_v)\,\mathrm{d}z, \tag{24}$$

In order to facilitate a comparison between the methods, we diagnose the number of used grid cells ($N$) for the adaptive-grid run. Figure 3a shows that for both approaches the error scales inversely proportional to the used number of grid cells to the second power (i.e. second-order spatial accuracy in 1D). The adaptive grid results are more accurate than the results from the

fixed-grid approach when employing the same number of grid cells. Figure 3b shows that for both approaches the required effort (i.e. measured here in wall-clock time) scales linearly with the number of grid cells, except for the runs that require less than one-tenth of a second to perform. The plots reveals that *per grid cell* there is computational overhead for the adaptive-grid approach. These results show that the used numerical solver is well behaved.

The following sections are devoted to testing the adaptive-grid approach in a more applied SCM scenario, where the turbulent transport closures are applied (see Sect. 2) and the set-up is unsteady. Here, the quality of the adaptive-grid solution has to be assessed by comparing against reference results from other SCMs, large-eddy simulations and the present model running in equidistant-grid mode.

## 3.2    GABLS1

The first GABLS intercomparison case focusses on the representation of a stable boundary layer. It's scenario was inspired by the LES study of an ABL over the Arctic sea by Kosović and Curry (2000). The results from the participating SCMs are summarized and discussed in Cuxart et al. (2006), for the LES intercomparison study, the reader is referred to the work of Beare et al. (2006). The case prescribes the initial profiles for wind and temperature, a constant forcing for momentum corresponding to a geostrophic wind vector, $U_{geo} = \{8,0\}$m/s and Coriolis parameter $f = 1.39 \times 10^{-4}$s$^{-1}$. Furthermore, a
fixed surface-cooling rate of $0.25$ K/hour is applied and $\theta_{v,ref} = 263.5K$. The model is run with a maximum resolution of $6.25$ meter and a domain height of $400$ meters. The maximum resolution corresponds to 6 levels of tree-grid refinement, where each possible coarser level corresponds to a factor of two increase in grid size.

Due to the idealizations in the case set-up with respect to the reality of the field observations, the model results were not compared against the experimental data (Cuxart et al., 2006). However, for the SCMs, a reference was found in the high-
fidelity LES results that tended to agree well between the various models. The LES results therefore serve as a benchmark for the results obtained with the present model. This facilitates a straightforward testing of the formulations and implementations of the used physical closures, before we continue our analysis towards the full diurnal cycle. Inspired by the analysis of Cuxart et al. (2006) and *their* figure 3, we compare our SCM results with the 6.25 meter-resolution LES ensemble results. We focus on the profiles for the wind components and potential temperature averaged over the ninth hour of the simulation in Fig. 4.
We observe that the present SCM is in good agreement with the LES results and is able to capture the vertical structure of the ABL, including the low-level jet. The differences are only minor compared to the variations in the results presented in the aforementioned GABLS1 SCM reference paper.

Note that in general, results are of course sensitive to the closure chosen to parameterize the turbulent transport, in our case given by Eqs.6 and 11. In order to separate between the numerical effects of using grid adaptivity and the chosen physical
closures, we define an additional reference case in which we run an *equidistant-grid* SCM. This model run employs a fixed 6.25 meter resolution (i.e with 64 cells), but otherwise identical closures and numerical formulations. I.e. we have switched-off the grid adaptivity and maintain the maximum resolution throughout the domain. We can observe that results between both SCMs are in good agreement but that minor deviations are present. These discrepancies are on the order of magnitude of the refinement criteria and can be reduced by choosing more stringent values, that would result in using more grid cells. The evolution of the

adaptive-grid structure is shown in Fig. 5 a. We see that a relatively high resolution is employed near the surface, i.e. in the logarithmic layer. Remarkably, without any a priori knowledge, the grid is refined at a height of $150 \,\mathrm{m} < z < 200 \,\mathrm{m}$ as the so-called low-level jet develops, whereas the grid has remained coarse above the boundary layer where the grid resolution was reduced to be as coarse as $100$ meters. From Fig.5 b we learn that the number of grid cells varied between $11$ and $24$ over the
course of the simulation run.

### 3.3  GABLS2

The second GABLS model intercomparison case was designed to study the model representation of the ABL over the course of two consecutive diurnal cycles. The case is set-up after the observations that were collected on the 23rd and 24th of October, 1999 during the CASES-99 field experiment in Leon, Kansas, USA (Poulos et al., 2002). The case prescribes idealized forcings
for two consecutive days that were characterized by a strong diurnal cycle pattern. During these days, the ABL was relatively dry, there were few clouds and $\theta_{v,ref} = 283.15 K$. The details of the case are described in the work of Svensson et al. (2011) that was dedicated to the evaluation of the SCM results for the GABLS2 intercomparison. Compared to the original case prescriptions, we choose a slightly higher domain size of $z_{top} = 4096$ meters (compared to $4000$ m), so that a maximum resolution of $8$ meters corresponds to $9$ levels of refinement.
In this section we place our model output in the context of the results presented in the work of Svensson et al. (2011), that, apart from the SCM results, also includes the results from the LES by Kumar et al. (2010). To obtain their data we have used the so-called 'data digitizer' of Rohatgi (2018). Inspired by the analysis of Svensson et al. (2011) and *their* figures 10 and 11, we compare our results for the wind-speed magnitude ($U = \|\boldsymbol{u}\|$) and virtual potential temperature profiles at 14:00 local time on the 23rd of October in Fig. 6 a and b, respectively. Here we see that the results obtained with the present SCM fall within
the range of the results as were found with the selected models that participated in the original intercomparison. These models also employed a first-order-style turbulence closure and have a lowest model-level height of less than $5$ meters. The present modelled virtual potential temperature ($\theta_v$) shows a slight negative vertical gradient in the well mixed layer. This is a feature related to the usage of the local $K$-diffusion description for the turbulent transport (see Sect. 2 and the work of Holtslag and Boville (1993)). Figure 6 c presents a timeseries of the 10-meter wind speed ($U_{10m}$) during the 23-rd of October. Again the
present model results compare well with the others. Next, in order to validate the grid-adaptivity independently from the used closures, we present the hourly evolution of the wind speed on the 24-th of October against the results obtained with adaptivity switched off, using 512 *equally-spaced* grid points in Fig. 7. A nearly identical evolution of the wind speed profiles is observed and even the small-scale features in the inversion layer (i.e. $z \approx 800$ m) are present in the adaptive-grid-model calculations. The corresponding evolution of the adaptive-grid structure is presented in Fig. 8, where the colours in the resolution plot
appear to sketch a 'Stullian' image, showing a prototypical diurnal evolution of the ABL (see figure 1.7 in the book of Stull, 1988). Apparently, the grid-adaptation algorithm has identified (!) the 'surface layer' within the convective boundary layer, the stable boundary layer, the entrainment zone and the inversion layer as the dynamic regions that require a high-resolution mesh. Conversely, the well-mixed layer within the CBL, the residual layer and the free-troposphere are evaluated on a coarser mesh. The total number of grid cells varied between $24$ and $44$.

## 4 Discussion & Conclusions

In this work we have presented a one-dimensional (1D) single-column model (SCM) that employs a mesh whose resolution is varied adaptively based on the evolution of the numerically obtained solution. This is an attractive feature because it is a prerequisite to enable the computational effort required for the evaluation of numerical solution to scale with the complexity of the studied physical system. The adaptation algorithm based on limiting discretization errors appears to function very well for a wide variety of geophysical applications: e.g. 3D atmospheric turbulence-resolving models (Van Hooft et al., 2018), tsunami and ocean-wave modelling (Popinet, 2011; Beetham et al., 2016; Marivela-Colmenarejo, 2017), hydrology (Kirstetter et al., 2016), two-phase micro physics (Howland et al., 2016), flow of granular media (Zhou et al., 2017) and shock-wave formation (Eggers et al., 2017). For these studies on highly dynamical systems, the adaptive-grid approach is chosen because it offers a more computationally efficient approach as compared to the usage of static grids.

The present work does not include an in-depth assessment and discussion on the performance of the presented methods in relation to the computational speed. Even though this is an important motivation for the application of the adaptive-grid strategy to GCMs, the authors argue that a SCM is not suitable for speed-performance testing: the speed of single-column calculations is virtually never a critical issue. Only in 3D mode, when SCMs are 'stitched together' to enable the resolving of global circulations, the model's computational efficiency becomes an issue. Furthermore, the performance of a SCM that employs a few tens of cells is not a good indicator for the performance of a GCM that can employ billions of grid cells. For the latter, parallel computation overhead and the so-called memory bottle neck are important aspects. In contrast, for the SCM case, the complete instruction set and solution data can typically be loaded onto the cache memory of a single CPU's core. Nevertheless, for the readers' reference, the required run times for the different SCM set-ups presented herein are listed in table 1, and figure 3b also presents quantitative results on this topic and shows that the adaptive-grid solver is well behaved.

Following the turbulence resolving study of Van Hooft et al. (2018), the results presented herein are a proof-of-concept for future 3D modelling, using RANS techniques. The authors of this work realize that the present SCM is a far cry from a complete global model and that more research and development is required before the method can be compared on a global-circulation scale. As shown by e.g. Jablonowski (2004), a 3D adaptive grid also allows a variable grid resolution in the horizontal directions. This further enables the computational resources to focus on the most challenging atmospheric processes where there is a temporal and spatial variation in the horizontal-resolution requirement of the grid. Examples include the contrasting dynamics between relatively calm centres of high-pressure circulations and those characterizing stormy low-pressure cells. Also, in the case of a sea breeze event (Arrillaga et al., 2016), it would be beneficial to temporarily increase the horizontal resolution near the land–sea interface. As such, we encourage the usage of this technique for those meteorologically challenging scenarios.

*Code and data availability.* Basilisk is a freely available (GPLv3), multi-purpose tool to solve partial differential equations and has it's own website: http://www.basilisk.fr. The code contains solvers for Saint-Venant problems, the Navier-Stokes equations, electro hydrodynamics and more, see http://basilisk.fr/src/README. A selection of examples can be viewed online: http://www.basilisk.fr/src/examples. The website also provides general information including; installation instructions and a tutorial. Furthermore, for the work herein, interested

readers can visit the model set-up pages and links to their online locations are presented in table 1. The data can easily be generated by running the scripts. Finally, a snapshot of the used code, as it was used in this the work, is made available via ZENODO, with doi link: https://doi.org/10.5281/zenodo.1203631.

*Author contributions.* All Authors contributed to the content of this manuscript and it should be viewed as a fruit of the many discussions
5   we have had. Furthermore, SP wrote the Basilisk code, he also designed and implemented the grid adaptation algorithm. The numerical experiments were set up and performed by AvH. The writing was led by AvH and organized in an iterative procedure with BvdW. Noting that the authors have no illusion they (can) do justice to the attribution of all the ideas and input.

*Competing interests.* The authors declare that there are no competing interests.

*Acknowledgements.* The authors gratefully acknowledge the funding by the ERC Consolidator grant (648666). The LES ensemble results
10  used for the GABLS1 intercomparison are kindly made available by Bob Beare; online via: http://gabls.metoffice.com/ and we thank three anonymous reviewers for their comments on the manuscript. Furthermore, we acknowledge that we are indebted to those who contributed to the GNU/Linux project and/or the GNU compiler collection (see www.gnu.org).

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

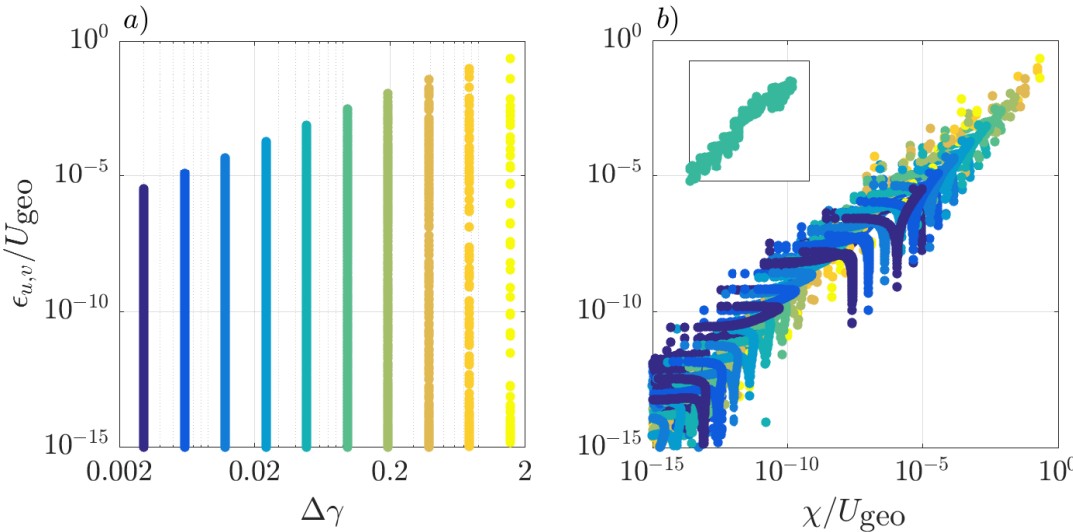

**Figure 1.** The locally evaluated error in the numerical solutions for $u$ and $v$ at $t = t_{end}$, based on the analytical solution ($\epsilon_{u,v}$, see Eq. 23) for 10 runs using different equidistant mesh-element sizes. The left-hand side plot (a) Shows that the diagnosed errors for each run plotted against the used mesh-element size ($\Delta$) times the inverse Ekman depth ($\gamma$, see text). The right-hand side plot (b) shows, with the same colour coding as in the left-hand side plot (a), the correlation between the wavelet-based estimated error ($\chi$) and the corresponding diagnosed error in the numerically obtained solution ($\epsilon$). The inset (using the same axis scales) shows the results for a single run, and reveal a spread of several orders of magnitude in both $\epsilon$ and $\chi$ values.

**Table 1.** The exact formulation of the methods are described at the online locations of the definition files for the different cases presented in this manuscript.

| Section | Case | Grid | URL: http://www.basilisk.fr... | Number of time steps | Wall-clock time |
|---|---|---|---|---|---|
| 3.1 | Ekman spiral | Adaptive | /sandbox/Antoonvh/ekman.c | 1000 ($\times$20 runs) | $\approx$ 19 sec. |
| " | " | Fixed & Equidistant | /sandbox/Antoonvh/ekmanfg.c | 1000 ($\times$10 runs) | $\approx$ 18 sec. |
| 3.2 | GABLS1 | Adaptive | /sandbox/Antoonvh/GABLS1.c | 16204 | $\approx$ 1.4 sec. |
| " | " | Fixed & Equidistant | /sandbox/Antoonvh/GABLS1fg.c | 16324 | $\approx$ 0.9 sec. |
| 3.3 | GABLS2 | Adaptive | /sandbox/Antoonvh/GABLS2.c | 24262 | $\approx$ 9 sec. |
| " | " | Fixed & Equidistant | /sandbox/Antoonvh/GABLS2fg.c | 33993 | $\approx$ 22 sec. |

The wall-clock times are evaluated using a single core (processor model: Intel i7-6700 HQ).

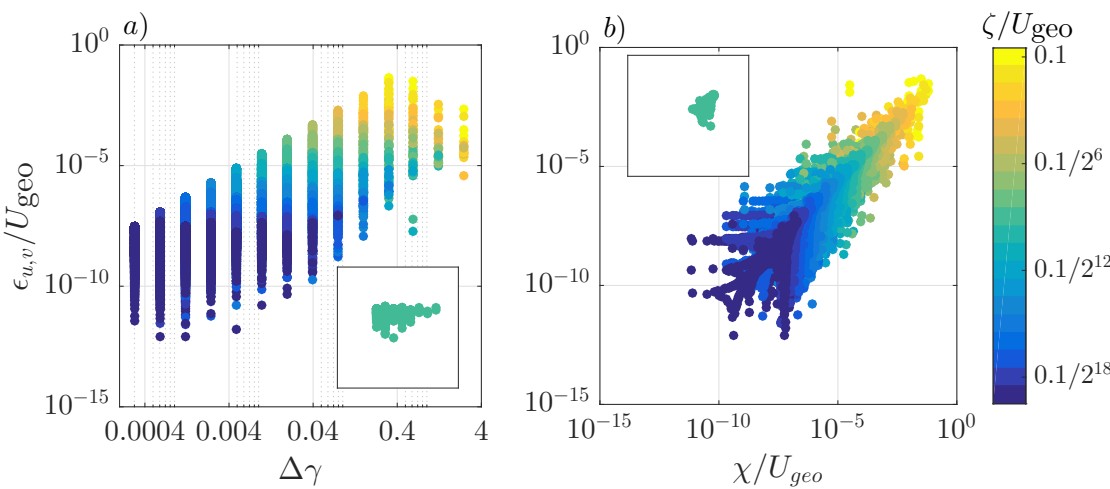

**Figure 2.** The locally evaluated error in the numerical solutions for $u$ and $v$ at $t = t_{end}$, based on the analytical solution ($\epsilon_{u,v}$, see Eq. 23) for 20 runs using the adaptive-grid approach with different refinement criteria (see colour bar). The left-hand side plot (a) Shows that the diagnosed errors for each run plotted against the used mesh-element size ($\Delta$). The inset (using the same axis scales) shows the results for a single run. The right-hand side plot (b) shows the correlation between the wavelet-based estimated error ($\chi$) and the corresponding diagnosed error in the numerically obtained solution ($\epsilon$). The inset (using the same axis scales) shows the results for a single run, and reveals a relatively small spread in both $\epsilon$ and $\chi$ values compared to the equidistant-grid results presented in figure 1b.

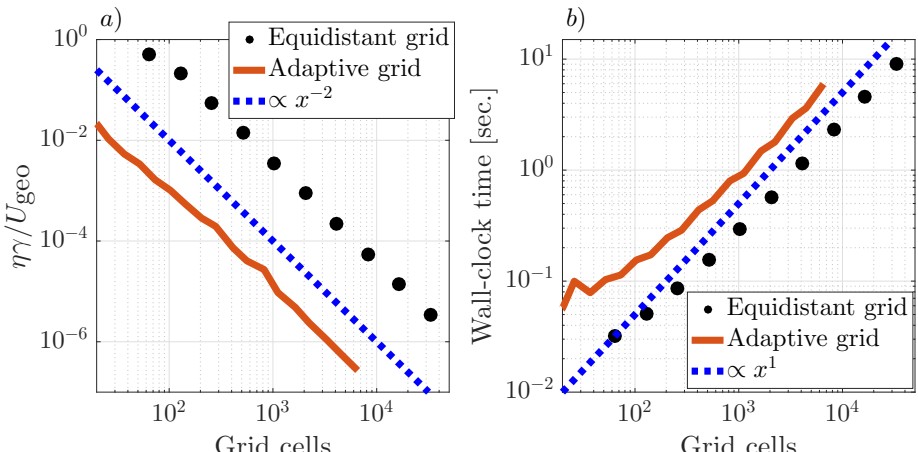

**Figure 3.** The scaling characteristics for the laminar-Ekman-spiral case. (a) Presents the error convergence for the equidistant-grid and adaptive-grid approach. The errors ($\eta$) follow the slope of the blue dashed line that indicates the second-order accuracy of the methods. The wall-clock time for the different runs is presented in (b), showing that for both of the aforementioned approaches, the required effort scales linearly with the number of grid cells.

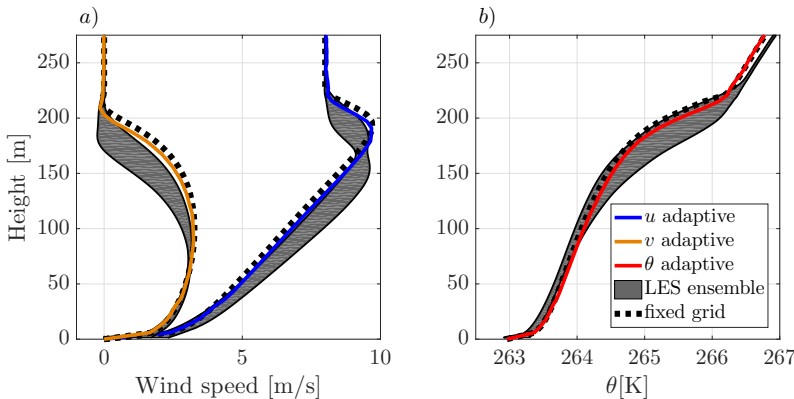

**Figure 4.** Time averaged profiles over the ninth hour of the run according to the GABLS1 intercomparison scenario. For (a) the horizontal wind components and (b) the potential temperature. Results are obtained with the present adaptive-grid SCM (coloured lines), the LES models ensemble (i.e mean $\pm\,\sigma$) from Beare et al. (2006) (grey-shaded areas) and the present SCM, employing an equidistant and static grid with a 6.25 meter resolution (dashed lines). For $z > 250$m, the profiles have remained as they were initialized.

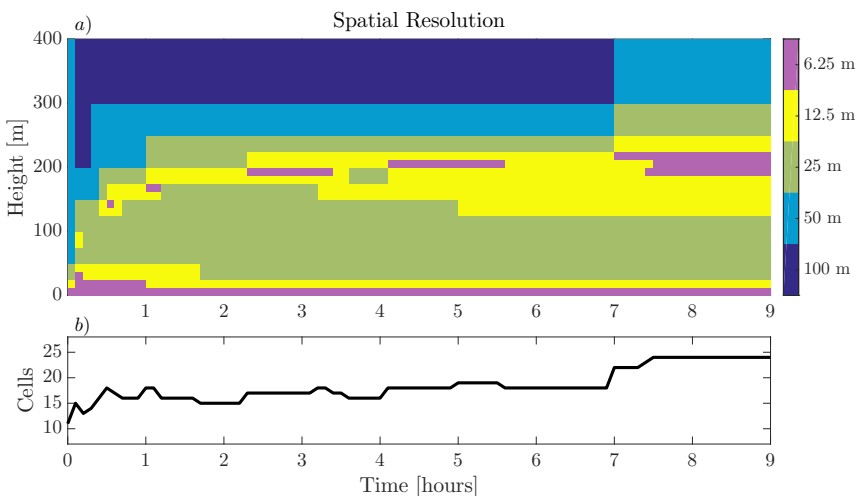

**Figure 5.** Evolution of (a) the vertical spatial-resolution distribution and (b) the total number of grid cells, for the GABLS1 intercomparison case.

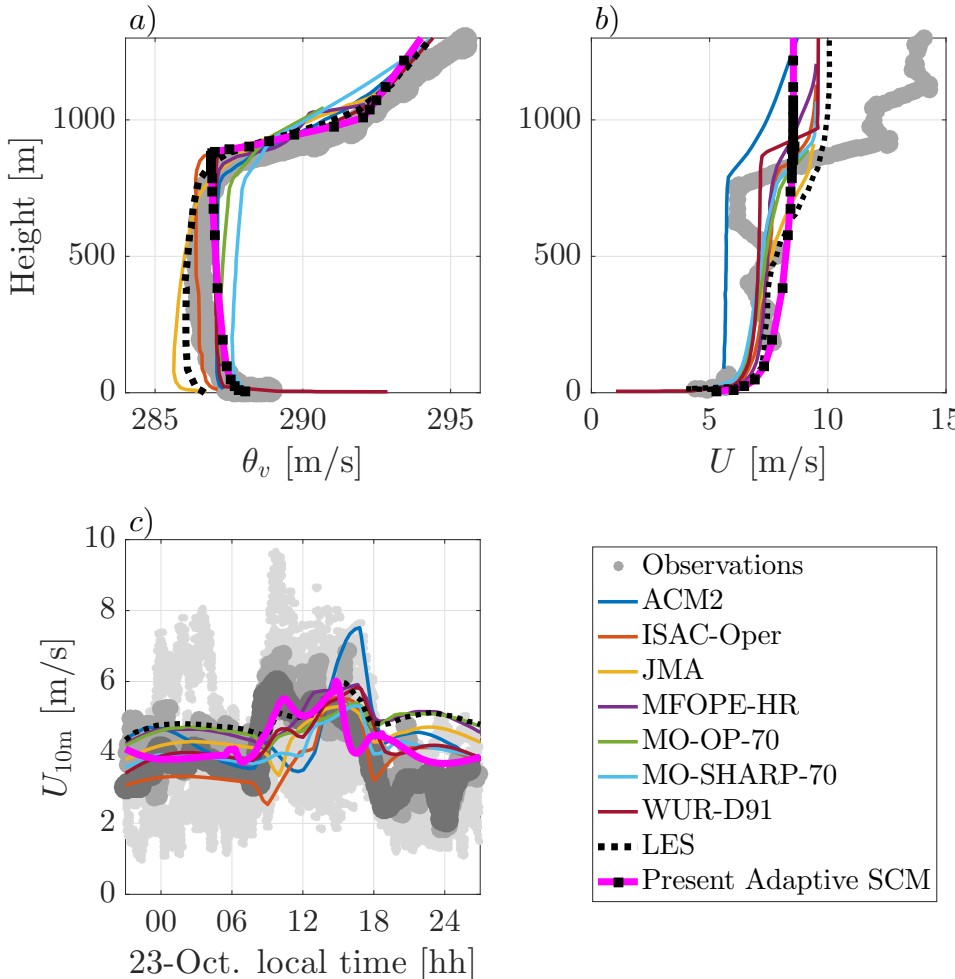

**Figure 6.** Comparison of the results obtained with the adaptive-grid SCM and the participating models in the work of (Svensson et al., 2011) for the vertical profiles of (a) the virtual potential temperature and (b) the wind-speed magnitude, for 14:00 local time on the 23rd of October. Lower panel: (c) the evolution of the 10-meter wind speed ($U_{10m}$) on the 23rd of October. For the used model abbreviations in the legend, see Svensson et al. (2011). The different shades of grey in plot c) indicate observations from different measurement devices, see Svensson et al. (2011) for the details.

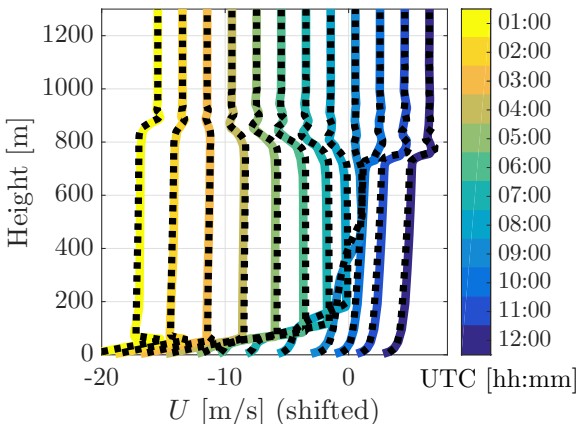

**Figure 7.** Vertical profiles of the wind-speed magnitude $U$ obtained with the adaptive-grid (in colour) and the fixed equidistant-grid (dashed) SCMs. The twelve plotted profiles are obtained for the 24th of October with an hourly interval, starting from 1:00 AM local time. Noting that the profiles are shifted in order to distinguish between the different times (with vanishing wind at the surface). The profiles of $U$ are constant with height for $z > 1200$m.

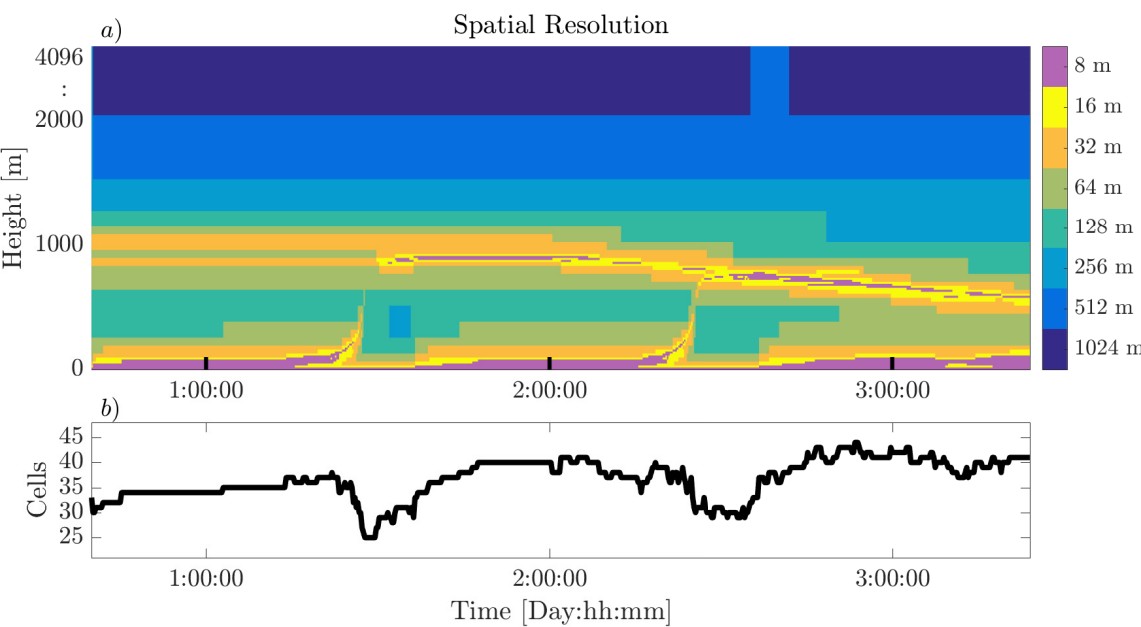

**Figure 8.** Evolution of (a) the vertical spatial resolution and (b) the total number of grid cells, for the GABLS2 intercomparison case. Two full diurnal cycles, corresponding to the 23rd and 24th day of October, 1999 (ranging from the labels 1:00:00 to 3:00:00 on the x-axis).