# Peer review of "Adaptive Cartesian Meshes for Atmospheric Single-Column Models, a study using Basilisk 18-02-16"

_Geoscientific Model Development, 2018_

## Referee Comment (RC1) · Anonymous Referee #1 · 27 Apr 2018

This article describes tests of an adaptive grid scheme in a single-column model for two ABL cases. The results are compared with a fixed-resolution version of the same model, and with various other models.

This is an interesting study and the results are clearly presented. The quality of the adaptive scheme solutions is encouraging. While the overall scope of the research is limited, it presents a possible avenue for future adaptive GCM development. There are a few places where more detail and clarification would be helpful (described below). I recommend acceptance of this article pending minor revisions.

In the model overview, it is stated that the grid refinement criteria are tuned based on

trial and error. How sensitive is the scheme performance to the tuning? If this type of adaptive scheme is implemented in a full GCM, will different tunings be necessary at different heights, geographical regions, or seasons?

Specific comments:

p.2 Sec. 1, Line 17: Sentence starting "However, it is important..." is unclear and needs to be rewritten

p.2 Sec. 1 Line 23: Sentence starting "This work departs..." makes it sound like this work uses different methods than van Hooft (2018), yet the next page Line 25 suggests the opposite. This sentence needs to be modified to make the meaning clearer.

p.3 Sec. 2 Line 12: Could you state at least the nature of the surface fluxes parameterization (e.g. bulk flux). Which type of closures in Holtslag and Boville are you referring to?

p.56 Sec. 3.2 : Could you add a little more qualitative description of this GABLS case? Were there clouds? Is it a surface driven convective BL? Is the wind shear significant or important?

p.5 Sec. 3 Line 12: It would be good to reiterate here that each 'level of refinement' halves the local grid spacing.

p.5 Sec. 3 Line 21: Are you saying the differences are only minor compared the the LES spread, or only minor compared to the SCM model spread?

Figure 3a: Unlike most models, thetav in your solution has a negative slope in the boundary layer, more negative even than the one other plotted model with a negative slope. Is this slope also consistent with your fixed-grid solution? Is there something atypical about your SCM physics that would allow this?

Figure 3c: What is the difference between the various shades of gray in the figure?

Figure 5 should have the simulation dates somewhere on the x-axis or at least the time

coordinates referenced to a date in the caption.

p. 6 Sec.3.2 Line 20: Stating that the evolution of the wind speed profile 'is the same' suggests that it is identical which is inaccurate. Perhaps 'is nearly the same'.

p. 6 Sec.3.2 Line 22: Could you clarify what is meant by 'Stullian image'? It would be helpful in this discussion if you qualitatively describe which parts of the diurnal cycle require the most/least refinement.

Technical corrections:

p.1 Sec. 1 Line 3: 'receives' should be 'receive'

p.2 Sec.2 Line 18: 'have' should be 'has'

p.4 Sec.2 Line 10: 'trail' should be 'trial'

p.4 Sec.2 Line 28: is 'that' referring to equation 6?

p.5 Sec.2 Line 33: Would be clearer as 'Online links are provied in table 1.'

p.5 Sec.3 Line 23: 'Noting' should be 'Note'

p.6 Sec.3.2 Line 17: '22-th' should be '22nd'

p.6 Sec.3.2 Line 22: 'Fig. 2' should be 'Fig. 5'

p.7 Sec.4 Line 3: 'efficient' misspelled

Appendix: Line 13: Sentence starting 'Using a domain...' is not a complete sentence.

Appendix: Line 18: 'facilitate' misspelled

Appendix: Line 18: would be clearer as 'we diagnose the number of used cells...'

Appendix: Line 19: Sentence starting 'Were the adaptive grid results....' is not a complete sentence.

Appendix: Line 22: Should be 'This plot reveals' or 'These plots reveal'

---

## Referee Comment (RC2) · Anonymous Referee #2 · 28 Apr 2018

A Basilisk 18-02-16 based adaptive grid scheme is employed and compared with an equal-distant high vertical resolution grid scheme in the same single-column atmospheric model for two land atmospheric boundary layer case studies. The diurnal variations of fine vertical structure near the bottom and the top of boundary layer is well captured using the adaptive grid scheme. Results are encouraging and clearly presented, which shows potential for future applications in global climate models. However the following major concerns are suggested to be addressed before acceptance for publication:

[1] In current state-of-art SCM/GCMs, more than 20s or more variables are involved

in physical and chemical process simulations. It is necessary to state clearly the basic rule for selecting the refinement criteria and to show sensitivity test results. For example in this study, the refinement criteria are assigned only for winds and temperature. The specific humidity Q is also a key physical variable in the SCM simulation, but no criteria is assigned, why? and how a new Q refinement critiria influences the scheme ghost points and overall cell points searching? And how a Q refinement critiria influences the boundary layer diurnal cycle (particularly the boundary layer clouds) simulation?

[2] In this study, the Basilisk 18-02-16 based adaptive grid scheme uses much shorter time-step (between "2 and 15 s" in page 4 line 29) than that of current state of art GCM/SCMs (which is around 10 to 20 minutes and vertical resolution is in the order of at least 100m). Considering Both radiation and vertical diffusion calculation is time consuming, using such a small time step will need much longer computing time. Is it possible to use normal time step of 10-20 minutes for the scheme? If yes, please add new time-step simulation results in Fig.1 to 5; if not, please discuss the limitations of the current adaptive scheme and propose a possible solution;

[3] In Fig. 3, the adaptive grid scheme simulated a slightly unstable (negative) virtual potential temperature profile above 100m while all other models simulate slightly stable (positive) profiles. Is it due to the adpative-grid scheme or the short-tail stability function used in the model or the Q profile difference,...? It is suggested to also add the fixed-resolution grid scheme results for comparison;

[4] Moist process is important in atmospheric boundary layer variations (both in diurnal and synoptical scale), to exam the effects of adaptive grid scheme on overall PBL simulation, vertical profile comparison of scheme simulated specific humidity Q is suggested to be added in Fig.1;

[5] Add diurnal cycle of observed and SCM simulated 2m temperature inter-comparison (similar like that of Fig.3 c for near surface wind speed) in Fig. 3 in order to better

understand the adaptive grid scheme performance.

---

## Referee Comment (RC3) · Anonymous Referee #3 · 24 May 2018

Review of the GMDD Manuscript: Adaptive Cartesian Meshes for Atmospheric Single-Column Models, a study using Basilisk 18-02-16

Authors: J. Antoon van Hooft, Stéphane Popinet, and Bas J.H. van de Wiel

The manuscript describes a prototype Single-Column Model (SCM) that employs dynamic grid adaptation in the vertical direction. The adaptations are guided by error estimators which increase the resolution where needed for accuracy, and remove grid points if additional accuracy is not needed. The method is tested in the atmospheric boundary layer based on the two GABLS intercomparison cases. In particular, the GABLS cases test the fine-scale diurnal variations of the planetary boundary layer (PBL). In general, the PBL structures in the two test cases are reasonably well captured in the adaptive SCM. The authors conclude that the adaptive-grid algorithm is able to dynamically coarsen and refine the numerical grid whilst maintaining an accurate solution.

Overall, the manuscript is interesting and well written, but the application area is rather narrow and it is difficult to judge the performance of the model based on two test cases. Therefore, it is difficult to draw more general conclusions concerning 3D General Circulation Models (GCMs). However, this is not the focus of this manuscript, which only addresses a prototype 1D adaptive model. The main criticism is that the description of the adaptive method is rather short which makes it difficult to understand the methodology even at a fundamental level. The main methodology is described in Van Hooft et al. (Boundary-Layer Meteorol., 2018) and the reader is referred to this paper. I recommend expanding the description of the Adaptive Mesh Refinement (AMR) algorithm somewhat in this paper, especially with respect to the error estimation technique. This will help make this a stand-alone paper. In addition, this review lists some clarifying questions that need to be addressed in a revised version.

**Specific comments:**
Page 2, line 23: In which way does your work depart from the work by Van Hooft et al. (2018)?
Page 3, line 17: Define the Richardson number. There are many variants, so please provide the equation.
Page 4, lines 5-10: Provide more details how the error estimator works. What makes it second-order-accurate?
Page 4, line 12 onwards: Clarifying questions concerning the refinements:
   1) Is the grid only refined if both error criteria are fulfilled, or is it enough if one error indicator is flagged?
   2) How often do you adapt (e.g. every time step)?
   3) How are newly-created grid points initialized, and how are coarsened grid points merged?
   4) Is the initialization (interpolation, merging) algorithm mass-conserving with respect to dry air mass & water mass and/or energy conserving?
   5) Does the interpolation/merging technique observe the hydrostatic balance? If not, one might expect a lot of gravity waves in 3D versions of this algorithm.
   6) Do you interpolate with respect to a height or pressure coordinate in the vertical direction?
   7) What is the order of the interpolation technique? If linear, are oscillation-free high-order

interpolation techniques available?

8) How many ghost cells are used?
9) Typical GCMs work with stretched grids and not equidistant grids. Can your algorithm be applied to stretched grids? The algorithm seems to rely on the fact that the grid spacing differs by exactly a factor of 2 (also refers to Fig. 2).
10) Have other error estimators (variables) and error thresholds been tried? If yes, comment on the pros and cons of these alternative choices.

Page 4, Eqs. (4)-(6): Since all operators are 1D, I suggest using partial derivatives with respect to z instead of the Laplacian operator in Eqs. (4) – (6)

Page 4, line 29: The time steps are extremely short. Can the adaptive method also be applied to more usual physics time steps on the order of minutes to half an hour?

Page 5, line 10: the geostrophic flow cannot be described as a 'forcing' mechanism. A forcing term needs units of $m/s^2$.

Page 5, line 17: should read 'physical closures for K'

Page 5, line 19: is this the potential temperature or virtual potential temperature? Fig. 1 shows the potential temperature, Fig. 3 show the virtual potential temperature. Is this difference intentional (provide some reasoning)?

Page 5, line 23: should read 'turbulent transport coefficient'

Page 5, line 28: should read 'on the order'

Page 6, Fig. 3: explain the meaning of the gray shading

Page 6, line 22: should read 'Fig. 5'

Page 6, line 27, should read 'presented a one-dimensional'

Page 7, line 15: explain acronym RANS

Page 8, Appendix: what is the equation that is solved here?

Page 8, line 10: provide the values for gamma, $U_{geo}$, the Coriolis parameter f, and the density rho

Page 8, line 19: Start sentence with 'The adaptive' instead of 'Were'.

Page 10, line 32: Update the Van Hooft et al. reference

Page 14, Table 1: State 'Number of time steps' instead of 'Solver timesteps'

Page 14, caption of Fig. A1: explain that the slope of the dashed line shows the second-order accuracy

---

## Author Comment (AC1) · 21 Jun 2018

**Response to Reviewer #1**

Antoon van Hooft, Stéphane Popinet, and Bas van de Wiel

June 2018

The authors thank the reviewer for taking his/her time to comment on the manuscript.

**The article describes tests of an adaptive grid scheme in a single-column model for two ABL cases. The results are compared with a fixed-resolution version of the same model, and with various other models. This is an interesting study and the results are clearly presented. The quality of the adaptive scheme solutions is encouraging. While the overall scope of the research is limited, it presents a possible avenue for future adaptive GCM development. There are a few places where more detail and clarification would be helpful (described below). I recommend acceptance of this article pending minor revisions.**

The authors agree with most points brought forward by the reviewer and we have therefore revised the manuscript accordingly. We hope that the manuscript is now more clear, and a point-by-point response is presented below. Also a PDF highlighting all the changes that were made with respect to the original manuscript is available.

> **In the model overview, it is stated that the grid refinement criteria are tuned based on trial and error. How sensitive is the scheme performance to the tuning? If this type of adaptive scheme is implemented in a full GCM, will different tunings be necessary at different heights, geographical regions, or seasons?**

It is the authors opinion that a complete discussion on the meaning, interpretation and selection of the refinement criterion warrants a study of its own. This is in fact part of our continued research, and hence is considered to be outside the scope of the present work. However, we agree that it is an important part of the grid adaptation algorithm that forms the basis of the present work. Therefore we have revised the manuscript considerably and have extended the analysis that was formerly in the appendix and moved it to the main text. Here we share our views on the usage of the criterion and argue that is provides a convenient and consistent framework for finding a balance between accuracy and performance. The new figures also provide quantitative results on this topic. How to translate these results obtained for the Ekman-spiral case to an SCM/GCM is still not obvious.

Furthermore, at this moment we cannot give a conclusive answer to the question regarding a variable refinement criterion. The authors do not see a good reason to employ a variable refinement criterion in time and space. Such an approach would mean that similar processes would be resolved with different accuracy depending on their localization in space and time. Our (current) philosophy is that for consistent meshing a pre-defined wish for accuracy can and should dictate the mesh's resolution depending of the resolved physics. On the otherhand, the physical processes that are dominant for the statistics of interest may change, depending on the time and location. Then, in practice, the accuracy requirements may vary and this could be refelected in the refinement criterion.

> **p.2 Sec. 1, Line 17: Sentence starting "However, it is important..." is unclear and needs to be rewritten.**

We hope the section on the concept of fractal scaling is more clear in the revised manuscript.

> **p.2 Sec. 1 Line 23: Sentence starting "This work departs..." makes it sound like this work uses different methods than Van Hooft (2018), yet the next page Line 25 suggests the opposite. This sentence needs to be modified to make the meaning clearer.**

We hope the sentence in clear in the revised manuscript.

> **p.3 Sec. 2 Line 12: Could you state at least the nature of the surface fluxes parameter- ization (e.g. bulk flux). Which type of closures in Holtslag and Boville are you referring to?**

Based on the reviewers comment, we have added a detailed description of the used closures in section 2 of the revised manuscript.

> **p.56 Sec. 3.2 : Could you add a little more qualitative description of this GABLS case? Were there clouds? Is it a surface driven convective BL? Is the wind shear significant or important?**

We have added a short overview of general conditions as they are modelled by the GABLS2 scenario. (lines )

> **p.5 Sec. 3 Line 12: It would be good to reiterate here that each 'level of refine-ment' halves the local grid spacing.**

We have re-iterated that in the revised manuscript.

> **p.5 Sec. 3 Line 21: Are you saying the differences are only minor compared the the LES spread, or only minor compared to the SCM model spread?**

We have revised the sentence to be more clear. We hope to convey the message that our SCM results are relatively close to the LES-ensemble results compared to the SCM results presented by the participants pf the original GABLS1 SCM intercomparison. Taking the LES as the 'truth': Notice that the fidelity in our results is due to the good performing description of the mixing closure by England and McNider (1995) (eq. 1 / 5 and 10 , original/revised), for this particular case.

> **Figure 3a: Unlike most models, thetav in your solution has a negative slope in the boundary layer, more negative even than the one other plotted model with a negative slope. Is this slope also consistent with your fixed-grid solution? Is there something atypical about your SCM physics that would allow this?**

Yes, this is a known feature of the to the used eddy-viscosity closure. Using this local

description for turbulent transport, a gradient is always "needed" for vertical mixing. This is not a realistic feature and better closures that account for counter-gradient transport are available. We have added a remark on this feature in the revised manuscript. Next we show that this gradient is indeed a formulation-specific feature. Therefore, we re-plot our results alongside results from runs with a different value for the maximum mixing length ($l_{bl}$, see Eqs. 7 and 8 in the revised manuscript). The results are presented in the figure below (or see attached figure in case it that now appear in this document). The default value for $l_{bl}$ was suggested in the reference literature and the used closures for transport are not the topic of our study. The figure also shows that the slope is not due to the usage of the grid adaptation algorithm (as was suggested by reviewer #2). In fact, the results for $\theta_v$ are within $0.2K$, which is much smaller than the difference with other models (up to multiple Kelvins). The used scripts to obtained these results or available and presented online (www.basilisk.fr/sandbox/Antoonvh/GABLS2forrev1and2.c). We choose not to add this analysis to the main text as it does not really add something new to the results or change the overall analysis of the manuscript. As all reviewers already agree: the results are encouraging.

figure-1.pdf

**Figure 3c: What is the difference between the various shades of gray in the figure?**
They are associated with different measurement techniques (see Svensson et al. (2011). Based on the reviewers comment we have stated this in the caption of this figure in the revised manuscript. Note that the forcings of for the GABLS2 case are idealized, and hence accuracy with respect to the measurements is not necessarily to be expected.

**Figure 5 should have the simulation dates somewhere on the x-axis or at least the time coordinates referenced to a date in the caption.**
We have added the date of the observation that inspired this (idealized) test case in

the caption.

The reviewer is right and the sentence is revised accordingly in the new manuscript.

Since the conceptual evolution of a diurnal cycle of the ABL as presented in Stull (1991) (His fig. 1.7) is such a well known image, we hope to coin the term "Stullian image". This was done with permission of Roland Stull (private communication). The manuscript aims to describe qualitatively what parts of the ABL required grid refinement. This was infact the goal of bringing up the similarities with the image of Stull. Therefore, we have chosen to keep that discussion 'as is'.

**Technical corrections:**
**p.1 Sec. 1 Line 3: 'receives' should be 'receive'**
**p.2 Sec.2 Line 18: 'have' should be 'has'**
**p.4 Sec.2 Line 10: 'trail' should be 'trial'**
**p.4 Sec.2 Line 28: is 'that' referring to equation 6?**
**p.5 Sec.2 Line 33: Would be clearer as 'Online links are provied in table 1.'**
**p.5 Sec.3 Line 23: 'Noting' should be 'Note'**
**p.6 Sec.3.2 Line 17: '22-th' should be '22nd'**
**p.6 Sec.3.2 Line 22: 'Fig. 2' should be 'Fig. 5'**
**p.7 Sec.4 Line 3: 'efficient' misspelled**
**Appendix: Line 13: Sentence starting 'Using a domain...' is not a complete sentence.**
**Appendix: Line 18: 'facilitate' misspelled**
**Appendix: Line 18: would be clearer as 'we diagnose the number of used cells...'**
**Appendix: Line 19: Sentence starting 'Were the adaptive grid results....' is not a com- plete sentence.**
**Appendix: Line 22: Should be 'This plot reveals' or 'These plots reveal'**

We thank the reviewer again for his/her careful reading of the manuscript and hope to have corrected the manuscript according to the suggested technical changes. Except for the suggestion that 'online' refers to the 'link (to)' rather than the location of the case set-up files.

**Supplement:**

---

## Author Comment (AC2) · 21 Jun 2018

Please see the Supplemented to review the proposed changes to the manuscript

Please also note the supplement to this comment:
https://www.geosci-model-dev-discuss.net/gmd-2018-21/gmd-2018-21-AC2-supplement.pdf

[Figure]

**Adaptive Cartesian Meshes for Atmospheric Single-Column Models, a study using Basilisk 18-02-16**

J.Antoon van Hooft[1], Stéphane Popinet[2], and Bas J.H. van de Wiel[1]

[1]Delft University of Technology, Department of Geoscience and Remote Sensing, Delft, The Netherlands
[2]Sorbonne Université, Centre National de Recherche Scientifique, UMR 7190, Institut Jean Le Rond D'Alembert, F-75005, Paris, France

*Correspondence to:* Antoon van Hooft (j.a.vanhooft@tudelft.nl)

**Abstract.** It is well known that the representation of certain atmospheric conditions in climate and weather models can still suffer from the limited grid resolution that is facilitated by modern-day computer systems. Herein we study a simple one-dimensional analogy to those models by using a Single-Column Model (SCM) description of the atmosphere. The model employs an adaptive Cartesian mesh that applies a high-resolution mesh only when and where it is required. The so-called

5   adaptive-grid model is described and we report on our findings obtained for tests to evaluate the representation of the atmospheric boundary layer, based on the first two GABLS intercomparison cases. The analysis shows that the adaptive-grid algorithm is indeed able to dynamically coarsen and refine the numerical grid whilst maintaining an accurate solution. This is an interesting result as in reality, transitional dynamics (e.g. due to the diurnal cycle or due to changing synoptic conditions) are rule rather than exception.

10   *Copyright statement.* All relevant rights reserved

**1   Introduction**

Single-Column Models (SCMs) are often used as the building blocks for Global (or General) Circulation Models (GCMs). As such, many of the lessons learned from SCM development can be inherited by GCMs and hence the evaluations of SCMs  receive considerable attention by the geoscientific model development community (see e.g. Neggers et al.,

15   2012; Bosveld et al., 2014; Baas et al., 2017). In this work, we present a SCM that employs an adaptive Cartesian mesh that can drastically reduce the computational costs of such models, especially when pushing the model's resolution. The philosophy is inspired by recently obtained results on the evolution of atmospheric turbulence in a daytime boundary layer using  three-dimensional (3D) adaptive grids. As promising results were obtained for  turbulence-resolving techniques such as Direct Numerical Simulations and Large-eddy Simulation (LES), herein we explore

20   whether similar advancements can be made with more practically oriented techniques for the numerical modelling of the atmosphere. As such, the present model uses Reynolds-averaged Navier-Stokes

**Fig. 1.**

---

## Author Comment (AC3) · 21 Jun 2018

**Response to Reviewer #2**

Antoon van Hooft, Stéphane Popinet, and Bas van de Wiel

June 2018

The authors thank the reviewer for taking his/her time to comment on the manuscript.

> **A Basilisk 18-02-16 based adaptive grid scheme is employed and compared with an equal-distant high vertical resolution grid scheme in the same single-column atmospheric model for two land atmospheric boundary layer case studies. The diurnal variations of fine vertical structure near the bottom and the top of boundary layer is well captured using the adaptive grid scheme. Results are encouraging and clearly presented, which shows potential for future applications in global climate models. However the following major concerns are suggested to be addressed before acceptance for publication:**

We are happy the reviewer find the results encouraging and hope to address the concerns in our point by point response and in the revised manuscript. A PDF highlighting the changes that were made is also provided as a supplement.

[1] In current state-of-art SCM/GCMs, more than 20s or more variables are involved in physical and chemical process simulations. It is necessary to state clearly the basic rule for selecting the refinement criteria and to show sensitivity test results. For example in this study, the refinement criteria are assigned only for winds and temperature. The specific humidity Q is also a key physical variable in the SCM simulation, but no criteria is assigned, why? and how a new Q refinement criteria influences the scheme ghost points and overall cell points searching? And how a Q refinement criteria influences the boundary layer diurnal cycle (particularly the boundary layer clouds) simulation?

The reviewer is right, finding a suitable mesh that strikes a balance between computational efficiency and accuracy of the diagnosed solution statistics is a challenge when performing numerical simulations. The challenge becomes even more prominent when statistics of over 20 variables need consideration. This is true for pre-tuned static anisotropic meshes, equidistant grids and we do not claim that the grid-adaptation algorithm lifts this burden from the model user either. Therefore, In the absence of a procedure for selecting mesh sizes for the static-grid approach (that typically also rely on trial-and-error, ad hoc testing and experience), the authors do not agree that the present (novel) approach should come with such guidelines. Especially when in practice, the grid (in)dependence of specific solution statistics is an arbitrary concept. More concretely: The SCMs in the original GABLS1 and GABLS2 intercomparison projects all use different meshes and similarly, a second (theorized) adaptive grid model could use different values for the refinement criteria to strike a different balance speed performance and numerical accuracy.

That said, the authors agree that the concept of the refinement criteria may raise new questions for modellers and currently warrants more study. Based on the reviewer's comments we made a serious effort to extend the analysis of the case that was formerly presented in the appendix (i.e. the laminar Ekman spiral), and we have included in it the main text. This new section (3.1), aims to exemplify for this simple case, what the effect is of tuning the refinement criterion, and argue that it provides a user-friendly,

convenient and consistent framework for finding a balance between computational efficiency and accuracy. Note that it is not obvious how these results would translate to an SCM and at this moment in time, we feel that the "trial-and-error approach" is an accurate description of how the current refinement criteria values for the SCM were found.

The reviewer is also right that it would be wise to extend the algorithm to also consider the atmospheric moisture content when pushing the method towards more realistic/applied scenarios. Note that this can be readily done by changing a line of code in the case set-up. However, for the GABLS1 case there is no moisture and for the GABLS2 case, moisture only slightly modifies the buoyancy and the location of its inversion corresponds to that of the inversion in temperature. Therefore, the present cases are too simple/specific to be suitable for finding a refinement criterion for the moisture content field. It is argued in the manuscript text that at this early stage within our developments/research, the simplicity of the cases is an advantage.

> **[2] In this study, the Basilisk 18-02-16 based adaptive grid scheme uses much shorter time-step (between "2 and 15 s" in page 4 line 29) than that of current state of art GCM/SCMs (which is around 10 to 20 minutes and vertical resolution is in the order of at least 100m). Considering Both radiation and vertical diffusion calculation is time consuming, using such a small time step will need much longer computing time. Is it possible to use normal time step of 10-20 minutes for the scheme? If yes, please add new time-step simulation results in Fig.1 to 5; if not, please discuss the limitations of the current adaptive scheme and propose a possible solution;**

The reviewer is right to bring forward this 'feature' of the present model. The text mentions that the time integration method is (only) first order accurate (page 4) and this limits the maximum time step of the present model. Also, the fact that the used resolution is much higher than the typical ABL resolution of an operational GCM, the temporal variations in the numerical solution contain contributions of higher frequencies, which warrant a reduced time step. The authors feel that

the original manuscript is clear and explicit on the fact that the present model is not an operational GCM. Rather, the the work aims reports on a possible avenue for an adaptive gridding strategy, that is also compatible with higher-order time-integration methods (see e.g. in the work of Rajarshi Roy Chowdhury online: http://basilisk.fr/sandbox/rajarshi/AllMach_O4/allmach_weno_poisson4_rk4.h). Therefore, the authors see no reason to expect that the adaptive grid strategy would be incompatible with the time integration strategies as they are currently used in GCM's.

Furthermore, when considering a process such as radiation for which it may be relatively expensive to calculate the corresponding tendency term but that is relatively slow in its evolution. The code also allows to only evaluate these tendencies every so often whilst the grid is able to adapt at intermediate time steps. This is exemplified here: www.basilisk.fr/sandbox/Antoonvh/smoke.c, and the results are published in earlier work of the present authors (Van Hooft et al. , 2018).

**[3] In Fig. 3, the adaptive grid scheme simulated a slightly unstable (negative) virtual potential temperature profile above 100m while all other models simulate slightly stable (positive) profiles. Is it due to the adaptive-grid scheme or the short-tail stability function used in the model or the Q profile difference,...? It is suggested to also add the fixed- resolution grid scheme results for comparison;**
The unstable profile is the results of the used $K$-closure. All of the other models from the intercomparison use a more recent (read: better) closure for their description of vertical mixing under unstable conditions (see text sect 2.). We have added a notion to this in the results section of the revised manuscript.

We have added a figure below (if it does not display, please see the supplement) that shows that the slope is controlled by the details of the used closure for turbulent mixing (i.e. the maximum mixing length $l$, see Sect. 2 of the revised manuscript). Also results for the default mixing length obtained with using the adaptive-grid and equidistant-grid approach are presented. It appears that the difference between both runs is small (max. $0.2K$) and that the slope is virtually identical. We choose not to add there results

to the revised manuscript as we feel that it does not add anything new to the present results or would change the analysis.

```
figure-1.pdf
```

**[4] Moist process is important in atmospheric boundary layer variations (both in diurnal and synoptical scale), to exam the effects of adaptive grid scheme on overall PBL simulation, vertical profile comparison of scheme simulated specific humidity Q is suggested to be added in the previous Fig.1;**
The reviewer is right that humidity plays an important role in the boundary dynamics. However, the GABLS1 case (corresponding to fig 1) does concern a dry boundary layer, and hence the results for the $q$ profiles are not presented.

**[5] Add diurnal cycle of observed and SCM simulated 2m temperature inter-comparison (similar like that of Fig.3 c for near surface wind speed) in Fig. 3 in order to better understand the adaptive grid scheme performance**
The maximum resolution in this simulation is 8 meters and the temperature at the surface is prescribed by the case definition. Therefore, the suggested statistic is not expected to be very sensitive to the used grid structure, but rather be a test of the used interpolation strategy. Alltough it would be interesting to extend the analysis, in all, the authors are confident that the current set of results supports the message we aim to convey sufficiently.

**Supplement:**

---

## Author Comment (AC4) · 21 Jun 2018

[revised manuscript text omitted]

The wall-clock times are evaluated using a single core (processor model: Intel i7-6700 HQ).

[Figure]

**Figure 2.** The locally evaluated error in the numerical solutions for $u$ and $v$ based on the analytical solution ($\epsilon_{u,v}$, see Eq. 19) for 20 runs using the adaptive-grid approach with different refinement criteria (see colourbar). The left-hand side plot (a) Shows that the diagnosed errors for each run plotted against the used mesh-element size ($\Delta$). The inset (using the same axis scales) shows the results for a single run. The right-hand side plot (b) shows the correlation between the wavelet-based estimated error ($\chi$) and the corresponding diagnosed error in the numerically obtained solution ($\epsilon$)). The inset (using the same axis scales) shows the results for a single run, and reveals a relatively small spread in both $\epsilon$ and $\chi$ values compared to the equidistant-grid results presented in figure 1b.

[Figure]

**Figure 3.** The scaling characteristics for the laminar-Ekman-spiral case. (a) Presents the error convergence for the equidistant-grid and adaptive-grid approach. The errors ($\eta$) follow the slope of the blue dashed line that indicates the second-order accuracy of the methods. The wall-clock time for the different runs is presented in (b), showing that for both of the aforementioned approaches, the required effort scales linearly with the number of grid cells.

[Figure]

**Figure 4.** Time averaged profiles over the eight hour of the run according to the GABLS1 intercomparison scenario. For (a) the horizontal wind components and (b) the potential temperature. Results are obtained with the present adaptive-grid SCM (coloured lines), the LES models ensemble (i.e mean $\pm \sigma$) from Beare et al. (2006) (grey-shaded areas) and the present SCM, employing an equidistant and static grid with a 6.25 meter resolution (dashed lines).

[Figure]

**Figure 5.** Evolution of (a) the vertical spatial-resolution distribution and (b) the total number of grid cells, for the GABLS1 intercomparison case.

[Figure]

**Figure 6.** Intercomparison of the results obtained with the adaptive-grid SCM and the participating models in the work of (Svensson et al., 2011) for the vertical profiles of (a) the virtual potential temperature and (b) the wind-speed magnitude. Lower panel: (c) the evolution of the 10-meter wind speed ($U_{10m}$) on the  23-th of October. For the used model abbreviations in the legend, see Svensson et al. (2011). The different shades of grey in plot c) indicate observations from different measurement devices, see Svensson et al. (2011) for the details.

[Figure]

**Figure 7.** Vertical profiles of the wind-speed magnitude $U$ obtained with the adaptive-grid (in colour) and the fixed equidistant-grid (dashed) SCMs. The twelve plotted profiles are obtained for the  24-th of October with an hourly interval, starting from 1:00 AM local time. Noting that the profiles are shifted in order to distinguish between the different times (with vanishing wind at the surface).

[Figure]

**Figure 8.** Evolution of (a) the vertical spatial resolution and (b) the total number of grid cells, for the GABLS2 intercomparison case. Two full diurnal cycles, corresponding to the 23-rd and 24-th day of October, 1999 (ranging from the labels 1:00:00 to 3:00:00 on the x-axis).

---

## Author Comment (AC5) · 21 Jun 2018

**Response to Reviewer #3**

Antoon van Hooft, Stéphane Popinet, and Bas van de Wiel

June 2018

The authors thank the reviewer for taking his/her time to comment on the manuscript. We hope to be able to address all points brought forward by the the reviewer in a point-by-point response in this document accompanied with relevant changes to the manuscript. A PDF highlighting all changes is also provided as a supplement.
The manuscript describes a prototype Single-Column Model (SCM) that employs dynamic grid adaptation in the vertical direction. The adaptations are guided by error estimators which increase the resolution where needed for accuracy, and remove grid points if additional accuracy is not needed. The method is tested in the atmospheric boundary layer based on the two GABLS intercomparison cases. In particular, the GABLS cases test the fine-scale diurnal variations of the planetary boundary layer (PBL). In general, the PBL structures in the two test cases are reasonably well captured in the adaptive SCM. The authors conclude that the adaptive-grid algorithm is able to dynamically coarsen and refine the numerical grid whilst maintaining an accurate solution.

Overall, the manuscript is interesting and well written, but the application area is rather narrow and it is difficult to judge the performance of the model based on two test cases. Therefore, it is difficult to draw more general conclusions concerning 3D General Circulation Models (GCMs). However, this is not the focus of this manuscript, which only addresses a prototype 1D adaptive model. The main criticism is that the description of the adaptive method is rather short which makes it difficult to understand the methodology even at a fundamental level. The main methodology is described in Van Hooft et al. (Boundary-Layer Meteorol., 2018) and the reader is referred to this paper. I recommend expanding the description of the Adaptive Mesh Refinement (AMR) algorithm somewhat in this paper, especially with respect to the error estimation technique. This will help make this a stand-alone paper. In addition, this review lists some clarifying questions that need to be addressed in a revised version.

Leaving out the the details of the error estimation technique was a choice that was made to prevent repetition of material that is published elsewhere. The referenced work in Bound.-Lay. Meteorol. (BLM) can be easily found, is freely available for everyone (open access CC 4.0) and in addition to that the work is also hosted via a mirror website at the HAL repository (https://hal.archives-ouvertes.fr/hal-01689036). This provides confidence that an interested reader will be able to find the more

detailed information if they wish so. The authors argue that repeating the 3-page story is therefore not necessary and does not really add to the proof-of-principle we aim to illustrate here. In the original manuscript we have opted to briefly describe the algorithm on a more conceptual level. In hindsight we agree that this may not suffice as it is indeed the key ingredient of the method. Therefore, based on the reviewers suggestion; a serious effort was made to include a didactical example based on an extension of the analysis of the laminar Ekman-spiral case that was formerly in the Appendix (in Sect 3.1). Now the manuscript includes a more detailed analysis of the usage of the error estimation technique. We feel that this is a valuable complementary example of the aforementioned work in the BLM paper, and not a repetition.

**Specific comments:**

**Page 2, line 23: In which way does your work depart from the work by Van Hooft et al. (2018)?**

Based on the point brought forward by the reviewer, we realize that we have not chosen our words careful enough here. The new manuscript is revised accordingly.

**Page 3, line 17: Define the Richardson number. There are many variants, so please provide the equation.**

The revised manuscript is now explicit on the used closures for turbulent transport and states all the relevant definitions.

[Figure]

**Page 4, line 12 onwards: Clarifying questions concerning the refinements:**
**1) Is the grid only refined if both error criteria are fulfilled, or is it enough if one error indicator is flagged?**
**2) How often do you adapt (e.g. every time step)?**
**3) How are newly-created grid points initialized, and how are coarsened grid points merged?**
**4) Is the initialization (interpolation, merging) algorithm mass-conserving with respect to dry air mass & water mass and/or energy conserving?**
**5) Does the interpolation/merging technique observe the hydrostatic balance? If not, one might expect a lot of gravity waves in 3D versions of this algorithm.**
**6) Do you interpolate with respect to a height or pressure coordinate in the vertical direction?**
**7) What is the order of the interpolation technique? If linear, are oscillation-free high-orderinterpolation techniques available?**
**8) How many ghost cells are used?**
**9) Typical GCMs work with stretched grids and not equidistant grids. Can your algorithm be applied to stretched grids? The algorithm seems to rely on the fact that the grid spacing differs by exactly a factor of 2 (also refers to Fig. 2).**
**10) Have other error estimators (variables) and error thresholds been tried? If yes, comment on the pros and cons of these alternative choices.**

The following answers in black are added to the main text, the answers in blue are not in the revised manuscript as they are considered off-topic for the present work:

1) If the estimated error in one (or more) of the three $(u, v, \theta_v)$ fields exceeds the respective criteria, the corresponding gridcell is refined.

2) The algorithm assesses the fidelity in the representation of the numerical solution at each time step, this garantees that no big developments in the solution take place in between grid adaptations. Noting that, courtesy of the tree-grid data structure, it is computationally cheap to do the assessment and refinement/coarsening compared to doing the time integration (i.e. typically less than 10% of the effort, for the presented

cases).

3) For refinement a bilinear interpolation technique is used whose second-order accuracy is consistent with the used solver. For coarsening, two cells can be merged into one by taking their average value which is exact for our finite-volume formulation.

4) The bilinear interpolation technique (for refinement) that is used in this study is not conserving for the first order moments of a scalar field, and not for higher order moments. However, the error introduced by this refinement step is directly controlled by the refinement criteria and can hence be tuned to any desired accuracy. Noting again that the second-order accuracy is inline with the solver's accuracy and hence is consistent with the overall method

5) Yes, see e.g. the 3D studies of van Hooft et al. (2018), or a more clean example online via the link: http://basilisk.fr/sandbox/Antoonvh/internalwavesAMR.c

6) No, In the model, height above the surface is used for this purpose instead

7) We use second order accurate bilinear formulation (using a 2-point stencil). Note that a conservative, linear interpolation technique based on a 3-point stencil is available and has an accuracy of the 3-rd order. Recent work of Radjarshi Roy Chowdhury does enable higher-order (of 3-rd, 4-th and 5-th order) methods that are non oscillatory. See an example test of his work that is based on so-called WENO schemes online: http://basilisk.fr/sandbox/rajarshi/WENO_CODES/weno_prolongation_scaling1D.c.

8) Two ghost cells are defined foreach resolution boundary and one at each "end" of the domain. This means that that in theory, for a worst-case-scenario, there may be as many ghost-cells as there are "solution grid cells". This is not really a concern since the values of the ghost cells are relatively cheap to calculate and only depend on the values of the "solution grid cells" that are solved for when time integrating the equations (typically $\mathcal{O}(10\%)$). Furthermore, figure 3 shows that the solver is well behaved.

9) The underlying (local) grid structure is Cartesian, and therefore any relevant mapping may be applied/implemented. (see e.g. http://basilisk.fr/src/README). However, the factor of two in the grid resolution between levels of refinement levels is an intrinsic

property of the tree-grid structure we use. Our results show that in an ABL the scale separation can differs by two-orders of magnitude within the GABLS2 domain (8m vs 1024m res.), meaning that the factor of two is not really an issue. Alternatively, adaptive unstructured grids exist that do not have this limitation (e.g. the code by the name of "fluidity" http://fluidityproject.github.io/). However, a complete discussion of the pro's and con's of such an approach would entail a new study of its own and is a considered to be outside the scope of the present work. Finally, the reference work of Dunbar (2008) uses dynamical grid stretching. But the authors would not call such a dynamic approach to be truly adaptive (see text).

10) Yes, this was part of the tria l-and-error approach and the pro's and con's may be obvious from the analysis in the new Sect. 3.1. The values of the refinement criteria may be used to tune the balance between accuracy and the speed performance of the code. Furthermore, for cases that are more driven by e.g. cloud-top radiation etc. it would be sensible to also refine based on the estimated errors in the moisture fields and cloud fraction field. Similar to pre-tuning a static grid, the balance between accuracy and speed performance remains at the discretion of the model user. The authors feel that the adaptation algorithm provides a more user-friendly, consistent and mathematically-rigorous approach compared to pre-tuning a stretched grid. This is especially true when the results from a model run are not know beforehand. Yet we cannot provide a universal recipe for finding suitable refinement-criterion values and this is part of our continued research.

> **Page 4, Eqs. (4)-(6): Since all operators are 1D, I suggest using partial derivatives with respect to z instead of the Laplacian operator in Eqs. (4) – (6)**

The equations are updated according to the reviewers suggestion

> **Page 4, line 29: The time steps are extremely short. Can the adaptive method also be applied to more usual physics time steps on the order of minutes to half an hour?**

The time-integration scheme is only first-order accurate and this hampers the time stepping. Not the adaptive grid scheme as we see a similar deterioration in our results for the fixed-grid solution when the time stepping parameter is relaxed to larger values.

Additionally, time stepping is small because the fine-scale features that are resolved, courtesy of the $O(10m)$ resolution. We note that there would be no reason to assume that adaptive-grids and/or the Basilisk code cannot handle the style of time-stepping schemes of operational GCM's. We argue that the manuscript is clear on the fact that we do not aim to present a GCM but we rather focus on a possible avenue for it's gridding.

**Page 5, line 10: the geostrophic flow cannot be described as a 'forcing' mechanism. A forcing term needs units of $m/s^2$ .**

The reviewer is right, the text has been improved accordingly in the revised manuscript.

**Page 5, line 19: is this the potential temperature or virtual potential temperature? Fig. 1 shows the potential temperature, Fig. 3 show the virtual potential temperature. Is this difference intentional (provide some reasoning)?**

We simply follow the original intercomparison papers regarding GABLS1 and GABLS2 of Cuxart et al. (2006) and Svensson et al. (2011), respectively. We think their reasoning was that the GABLS1 case is dry and hence the concept of virtual potential temperature is not relevant over the its non-virtual counterpart. This is not the case for GABLS2, where in the original work of Svensson et al. it was chosen to intercompare the virtual potential temperature.

**Page 5, line 17: should read 'physical closures for K'**
**Page 5, line 23: should read 'turbulent transport coefficient'**
**Page 5, line 28: should read 'on the order'**
**Page 6, Fig. 3: explain the meaning of the gray shading**
**Page 6, line 22: should read 'Fig. 5'**
**Page 6, line 27, should read 'presented a one-dimensional'**
**Page 7, line 15: explain acronym RANS**
**Page 8, Appendix: what is the equation that is solved here?**
**Page 8, line 10: provide the values for gamma, U geo , the Coriolis parameter f, and the density rho**
**Page 8, line 19: Start sentence with 'The adaptive' instead of 'Were'.**
**Page 10, line 32: Update the Van Hooft et al. reference**
**Page 14, Table 1: State 'Number of time steps' instead of 'Solver timesteps'**
**Page 14, caption of Fig. A1: explain that the slope of the dashed line shows the second-order accuracy**

We thank the reviewer again for his/her carefull reading of the manuscript and hope to have adressed all these points at their corresponding locations in the revised manuscript.

Except that we have not added the numerical values for $U_{geo}$, $\Omega$ etc... In stead, based on the reviewers comment, we present the results now in a properly scaled frame work. Making the results universal, as is allowed by the fact that the (scaled) analytical Ekman solution is not a function of the dimensionless ratio: $\Pi = \frac{U_{geo}\gamma}{\nu}$.

**Supplement:**

[revised manuscript text omitted]

The wall-clock times are evaluated using a single core (processor model: Intel i7-6700 HQ).

[Figure]

**Figure 2.** The locally evaluated error in the numerical solutions for $u$ and $v$ based on the analytical solution ($\epsilon_{u,v}$, see Eq. 19) for 20 runs using the adaptive-grid approach with different refinement criteria (see colourbar). The left-hand side plot (a) Shows that the diagnosed errors for each run plotted against the used mesh-element size ($\Delta$). The inset (using the same axis scales) shows the results for a single run. The right-hand side plot (b) shows the correlation between the wavelet-based estimated error ($\chi$) and the corresponding diagnosed error in the numerically obtained solution ($\epsilon$)). The inset (using the same axis scales) shows the results for a single run, and reveals a relatively small spread in both $\epsilon$ and $\chi$ values compared to the equidistant-grid results presented in figure 1b.

[Figure]

**Figure 3.** The scaling characteristics for the laminar-Ekman-spiral case. (a) Presents the error convergence for the equidistant-grid and adaptive-grid approach. The errors ($\eta$) follow the slope of the blue dashed line that indicates the second-order accuracy of the methods. The wall-clock time for the different runs is presented in (b), showing that for both of the aforementioned approaches, the required effort scales linearly with the number of grid cells.

[Figure]

**Figure 4.** Time averaged profiles over the eight hour of the run according to the GABLS1 intercomparison scenario. For (a) the horizontal wind components and (b) the potential temperature. Results are obtained with the present adaptive-grid SCM (coloured lines), the LES models ensemble (i.e mean $\pm \sigma$) from Beare et al. (2006) (grey-shaded areas) and the present SCM, employing an equidistant and static grid with a 6.25 meter resolution (dashed lines).

[Figure]

**Figure 5.** Evolution of (a) the vertical spatial-resolution distribution and (b) the total number of grid cells, for the GABLS1 intercomparison case.

[Figure]

**Figure 6.** Intercomparison of the results obtained with the adaptive-grid SCM and the participating models in the work of (Svensson et al., 2011) for the vertical profiles of (a) the virtual potential temperature and (b) the wind-speed magnitude. Lower panel: (c) the evolution of the 10-meter wind speed ($U_{10m}$) on the  23-th of October. For the used model abbreviations in the legend, see Svensson et al. (2011). The different shades of grey in plot c) indicate observations from different measurement devices, see Svensson et al. (2011) for the details.

[Figure]

**Figure 7.** Vertical profiles of the wind-speed magnitude $U$ obtained with the adaptive-grid (in colour) and the fixed equidistant-grid (dashed) SCMs. The twelve plotted profiles are obtained for the  24-th of October with an hourly interval, starting from 1:00 AM local time. Noting that the profiles are shifted in order to distinguish between the different times (with vanishing wind at the surface).

[Figure]

**Figure 8.** Evolution of (a) the vertical spatial resolution and (b) the total number of grid cells, for the GABLS2 intercomparison case. Two full diurnal cycles, corresponding to the 23-rd and 24-th day of October, 1999 (ranging from the labels 1:00:00 to 3:00:00 on the x-axis).

---

## Referee Report (RR1)

Review of the revised GMDD Manuscript: Adaptive Cartesian Meshes for Atmospheric Single-Column Models, a study using Basilisk 18-02-16

Authors: J. Antoon van Hooft, Stéphane Popinet, and Bas J.H. van de Wiel

**Comments from Reviewer 3:**
The revised manuscript has taken most of my previous comments and questions into consideration. However, the new manuscript has now raised many new questions and concerns, as there are several sign, math and physics errors in the new equations, inadequate descriptions of the test cases and forcing mechanisms, and missing parameter values and undefined symbols. The authors need to provide enough explanations to enable others to repeat the test setups.
In case the sign and math errors were present in the computations, all results need to be repeated and reevaluated. This might require major revisions.

**Detailed comments:**
1) Eqs. 1a-1d: Using references like Holtslag and Boville (1993), Liu et al. (Mon. Wea. Rev., Feb. 2013) or Andreas and Murphy (J. Physical Oceanography, Nov. 1986), all four surface flux equations 1a-1d have the wrong signs. Explain the sign discrepancy to the aforementioned papers and the sign convention used in this manuscript. In addition, the definition of $q_0$ needs to be 'saturation specific humidity at the surface'. There is a possibility that the surface fluxes have been incorrectly applied in this manuscript, which would necessitate a repetition of all simulations. It is also noted that the cited reference Louis (1982) for the surface fluxes does not exist. It is likely that the authors mean the paper Louis et al. (1982):
Louis, J.-F., Tiedtke, M, and Geleyn, J.-F.: A short history of the operational PBL parameterization at ECMWF. Proceedings of the Workshop on Planetary Boundary Layer Parameterization, 25-27 November 1981, ECMWF, Reading, U.K., 59-79, 1982
https://www.ecmwf.int/en/elibrary/10845-short-history-pbl-parameterization-ecmwf
However, this paper does contain any discussion of the surface fluxes (only some exchange coefficients) and is therefore an inadequate reference for the surface fluxes on page 3 line 14. The other provided reference Beljaars et al. (1989) is gray literature (is this an internal technical report, there is insufficient information), it is not available online, and has limited value here. Provide a better reference.

2) Eq. 3: There is again a sign error in this equation. The current formulation wrongly leads to a negative surface-layer bulk Richardson number for stable conditions with $\theta_{v,1} > \theta_{v,0}$. Such a stable stratification needs to have a positive $Ri_b$ number (see also Holtslag and Boville (1993), their Eq. (2.8), for the correct definition). Since $Ri_b$ is used in Eq. (5), there is the potential that most of the simulations in this manuscript are wrong. This needs to be clarified.

3) Page 4, lines 4 & 5, and Eq, (3): The definition of $\theta_{v,ref}$ is vague. What do you mean by 'reference value'? Provide the exact definition. Obviously, this reference value of $\theta_{v,ref}$ in

the surface layer must be different than the $\theta_{v,ref}$ values used later in the equation for the planetary boundary layer (Eq. (11)). However, the same symbol is used, and no further explanations are offered. Correct this.

4) Page 4, line 8-9: the capital $Z_{0,M}$ symbol is undefined, needs to be $z_{0,M}$. Provide the value of the roughness length to make the results reproducible.

5) Eq. (9): Incorrect definition of the vertical wind shear magnitude. It needs to read

$$S = \left| \frac{d\vec{v}}{dz} \right| = \left\| \begin{pmatrix} \dfrac{du}{dz} \\ \dfrac{dv}{dz} \end{pmatrix} \right\| = \sqrt{\left(\frac{du}{dz}\right)^2 + \left(\frac{dv}{dz}\right)^2}$$

instead of the currently used definition

$$S = \left\| \frac{dU}{dz} \right\| = \left\| \frac{d\left[\left(u^2 + v^2\right)^{1/2}\right]}{dz} \right\|$$

with undefined symbol $\| \; \|$. If the incorrect formulation has been used in the computations, they will need to be repeated.

6) Eqs. 14, 15, 16: You converted the former vector equation to a scalar equation (as requested) but left the scalar product operator in the formulation. This is mathematically incorrect for the scalar formulation. The dot product needs to be removed. At which time level is the forcing 'r' evaluated? Explain (page 6, line 24) that 'n' denotes the time level.

7) section 3.1: The description of the Ekman spiral test is insufficient and needs thorough revisions. It is furthermore unclear how it is correctly implemented.
I disagree with the author's reply to my first review that it is not necessary to know the values of the parameters. Without the given values of
    $U_{geo}$, f (and thereby the latitude angle $\phi$), $\Omega$, $\nu$, $\rho$
the test case is irreproducible. These values need to be provided. In addition, it needs to be clarified that
(a) $\nu$ is constant (hidden information via the words 'without any closures', is this correct?) and serves the role of $K(=\nu)$ in Eqs. 14-16
(b) Eqs. (1)-(11) are irrelevant for the discussion
(c) the exact definition for the forcing terms r needs to be provided for u and v.
In order to arrive at the analytical solutions (17) and (18) of the Ekman spiral, it must be assumed that the motion vanishes at z = 0 and tends to the zonal

geostrophic value $\vec{v} = \begin{pmatrix} U_{geo} \\ 0 \end{pmatrix}$ in the free atmosphere. In addition, the Ekman

solution (17) and (18) is based on the equation set

$$0 = K\frac{\partial^2 u}{\partial z^2} + fv - \frac{1}{\rho}\frac{\partial p}{\partial x}$$

$$0 = K\frac{\partial^2 v}{\partial z^2} - fu - \frac{1}{\rho}\frac{\partial p}{\partial y}$$

When comparing this formulation to Eq. (14) identify exactly how the forcing term r represents the forcing from the Coriolis and pressure gradient term in the u and v equations (provide the equations for $r_u$ and $r_v$).

Note that your definition of

$$\frac{dP}{dy} = U_{geo}f\rho \quad \text{(page 7, line 14)}$$

seems to have a sign error and might need to read $\frac{dP}{dy} = -U_{geo}f\rho$ if you imply a

geostrophic balance.

8) The other problem with section 3.1 is the incorrect definition of γ (line 20) which needs to be

$$\gamma = \sqrt{\frac{f}{2K}} = \sqrt{\frac{2\Omega\sin\phi}{2K}} = \sqrt{\frac{\Omega\sin\phi}{v}} .$$

Only in the very special case of φ=π/2 (North Pole) is this equation identical to the definition of γ in the manuscript. However, this is not specified, and γ might be used in an incorrect way. In addition, the authors call the quantity γ 'Ekman depth'. Since the physical units of γ are $m^{-1}$ this is inadequate (it is an inverse). The γ definition is then wrongly used in the definitions of $z_{top}$ (line 22), $t_{end}$ and dt. The physical units do not work out. Divisions by γ are needed instead of multiplications. The wrong use of γ also affects Figs. (1) and (2). The x-axis label Δ/γ must have units of $m^2$ in the current version (not dimensionless). Figs. 1 and 2 furthermore suggest that $U_{geo}$ = 1 m/s was selected in practice. This is the necessary value to represent the upper error limit of 0.25 m/s along the scaled $\varepsilon/U_{geo}$ y-axis in Figs. 1 and 2. Is this assumption correct? All these aspects need to be clarified.

9) Page 8, line 26: how do the 1000 time steps compare to the setting of $t_{end}$ and dt?

10) Section 3.2: Point out that this is a dry test case. It looks as if the GABLS1 case only forces the *zonal* momentum (line 19). Also add the information about the constant Coriolis parameter f and the density ρ. Does the density vary with height and if yes, how? As in section 3.1, provide the exact forcing functions $r_u$, $r_v$ and $r_\theta$. It seems clear how Eqs. (6)-(11) connect to Eqs. (14)-(16) (via the computation of K), but it is unclear how the surface flux equations (1)-(5) enter Eqs. (14)-(16). Provide this information.

Cuxart et al. (2006) presented their results after 9 hours (averaged over the 9th hour). You average the results over the 8th hour and compare to Cuxart et al. (2006). What is the reason for the discrepancy? Are the results converged enough to a

steady-state solution that the 8th and 9th hour time frames become comparable? Provide an explanation.

11) Page 10, line 25 and Figs. 6a,b: which time snapshot is shown? Add this information to the text and the figure caption.
The domain is 4000 m high, but only 1300 m are shown in Fig. 6? Why? How do the solutions compare in the upper domain?

**Correction of typos and style:**
Page 1, line 15: '… an SCM …'
Page 2, line 27: ' … built-in …'
Page 4, line 14: '.. description …'
Page 5, line 24 and page 7, line 25: '… its ….'
Page 6, line 17: '… spent …'
Page 7, line 10: should read ' … clean setup quantifies numerical errors explicitly and tests the  …'
Page 8, line 2: Bring footnote into the main text
Page 8, line 3: '…shows the results of the errors at all levels and …'
Page 8, line 15: '… though…'
Page 8, line 28: '… arise in the solution …'
Page 9, line 1: '…and the computational performance …'
Page 10, line 1: '… parameterize …'
Page 10, line 6: … on the order of …'
Page 11, line 2: 'Fig. 5' needs to read Fig. 8
Page 11, line 8: '… presented a one-dimensional …'
Caption Fig. 1 and 2: Add the information that the errors are shown at $t_{end}$ (the end of the simulation). Also add: the inset shows the errors for all time steps.
Fig. 3: symbol 'L' is undefined
Caption, Fig. 4: '… eighth hour …'. Do u and v stay constant above 275 m?

---

## Referee Report (RR2)

Review of the revised GMDD Manuscript: Adaptive Cartesian Meshes for Atmospheric Single-Column Models, a study using Basilisk 18-02-16

Authors: J. Antoon van Hooft, Stéphane Popinet, and Bas J.H. van de Wiel

**Comments from Reviewer 3 (3rd review):**
The newly revised manuscript (September 2018) has greatly improved, answered almost all of the questions from the 2nd review, and cleaned up most of the erroneous equations. However, two sign errors remain in the current version of the manuscript. As before, the authors need to check whether these sign errors were present in the calculations, or whether these are typos in the manuscript. In case the errors are in the code, the results need to be reproduced. I also have a few more clarifying questions, and suggest a few corrections of the text.

**Detailed comments:**

1) Page 4, lines 4 & 5, and Eq. (3) and page 5, Eq. (12): As I pointed out in my 2nd review the definition of $\theta_{v,ref}$ is vague. My earlier review asked what you meant by 'reference value'? The authors provided a satisfactory answer in their reply to my review comment, but none of this information made it into the manuscript. For example, the reply stated that a constant $\theta_{v,ref}$ value (however, it is not provided) is used in the ABL case, and that the GABLS tests define this reference profile. Therefore, the manuscript still lacks clarity. It makes it impossible to reproduce the ABL results (provide $\theta_{v,ref}$), and the authors should point to the GABLS tests for their specific $\theta_{v,ref}$ definitions. Please add the information from the reply to the manuscript.

2) Page 5, Eq. (13) and line 17: Do you refer to the same spatially- and time-dependent K coefficient that is defined in Eq. (8)?

3) Page 5, Eq. (15): The K diffusion coefficient as defined in Eq. (8) has a time dependency. Indicate in Eq. (15) whether K gets evaluated at time level n or the future time level n+1.

4) Page 6, line 5: the expression for the 'pressure gradient force vector' must be $-\dfrac{1}{\rho}\nabla p$, and not just the negative gradient $-\nabla p$ as currently displayed.

5) Page 6: I had already pointed out the sign error in the geostrophic relationship in my 2nd review, but these errors are still present. The right hand side of Eq. (17) must read $-\dfrac{1}{\rho}\nabla p - f\left(\vec{k}\times\vec{u}\right)$ and Eq. (18) must read $\vec{U}_{geo} = \dfrac{1}{\rho f}\vec{k}\times\nabla p$. If the sign errors are present in the code, the results need to be reproduced.

6) Page 6, line 9: It is incorrect to define the symbol $\times$ as the 'vector outer product operator' (which leads to a matrix). The symbol is the 'cross product' (leading to a vector).

7) Page 7, end of line 10: typo, should read '… of a choice …'

---

## Author Response (AR2)

**Response to the Reviewers**

Antoon van Hooft, Stéphane Popinet, and Bas van de Wiel

August 2018

The authors thank the reviewers for taking their time to comment on the revised manuscript. The comments of the reviewers are discussed point-by-point. After that, the revised manuscript is presented in a form where all changes made to the old revised manuscipt to arrive at the current version of the manuscript have been highlighted.

**1 Response to Reviewer #1**

We thank the reviewer for the suggested technical corrections.

> **Figure 8 highlights the diurnal cycles influence on grid refinement. The timing of local midnight or local noon should be noted somewhere.**
> **p.2 Line 17: would be clearer as '..., enabling better resolution of the most demanding processes.'**
> **p.3: $C_M$ and $C_H$ are undefined.**
> **p.8 Line 10: should be 'empiricism'**
> **p.8 Line 15: should be 'Even though'**
> **p.11 Line 2: shouldn't this be Figure 8 not Figure 5?**
> **Figure 1 caption last line: 'several'**
> **Figure 6 caption last line: 'measurement'**

We have revised the manuscript accordingly. Noting that we have not changed figure 8 as it already noted the timings of midnight. Furthermore, Eqs. 1a - d introduce $C_H$ and $C_M$, they are calculated using the definitions of Eqs. 2,3, 5 and 6. (i.e. 2,3,4,5 in the original revised manuscript).

**2 Response to Reviewer #2**

In the absence of any further comments, this section does not contain a reponse.

**3 Response to Reviewer #3**

We thank the reviewer for his/hers usefull comments on the revised manuscript. I certainly helped the authors to improve the quality of the work.

> **Comments from Reviewer 3: The revised manuscript has taken most of my previous comments and questions into consideration. However, the new manuscript has now raised many new questions and concerns, as there are several sign, math and physics errors in the new equations, inadequate descriptions of the test cases and forcing mechanisms, and missing parameter values and undefined symbols. The authors need to provide enough explanations to enable others to repeat the test setups. In case the sign and math errors were present in the computations, all results need to be repeated and reevaluated. This might require major revisions.**

The authors agree with most points brought forward by the reviewer and we have therefore revised the manuscript accordingly. We regret that the used closures for the turbulent transport that were added to the first revision were often not written down properly and we again thank the reviewer for his/her review and noticing these errors. It appears we have not been thorough enough when we casted the used formulations in their symbolic mathematical form. For all the issues brought forward regarding the equations were of typographical nature. The correct formulations were in fact already coded in our numerical solver and hence the results do not require an update and we have double checked this. We also hope to adress the other issues raised by the reviewer, both in the new revised manuscript and the point by point response presented below.
* * *
**1) Eqs. 1a-1d: Using references like Holtslag and Boville (1993), Liu et al. (Mon. Wea. Rev., Feb. 2013) or Andreas and Murphy (J. Physical Oceanography, Nov. 1986), all four surface flux equations 1a-1d have the wrong signs. Explain the sign discrepancy to the aforementioned papers and the sign convention used in this manuscript.**

The reviewer is right; we should follow the convection of upward fluxes and we have updated the corresponding equations.
* * *
**In addition, the definition of q 0 needs to be 'saturation specific humidity at the surface'. There is a possibility that the surface fluxes have been incorrectly applied in this manuscript, which would necessitate a repetition of all simulations.**

In General, the value of the specific humidity at the surface is not necessarily the saturation specific humidity. For some physical scenario's it would not be an accurate description (e.g. when the soil is drying). For this work we consider its prescription to be a case specific detail, for which we refer to the work of Cuxart et al. (2006) (were $q_0 = 0$) and Svensson et al. (2011). Indeed, for the GABLS2 case it is taken as the saturation specific humidity.
* * *
**It is also noted that the cited reference Louis (1982) for the surface fluxes does not exist. It is likely that the authors mean the paper Louis et al. (1982): Louis, J.-F., Tiedtke, M, and Geleyn, J.-F.: A short history of the operational PBL parameterization at ECMWF. Proceedings of the Workshop on Planetary Boundary Layer Parameterization, 25-27 November 1981, ECMWF, Reading, U.K., 59-79, 1982 https://www.ecmwf.int/en/elibrary/10845-short-history-pbl-parameterization- ecmwf However, this paper does contain any discussion of the surface fluxes (only some exchange coefficients) and is therefore an inadequate reference for the surface fluxes on page 3 line 14. The other provided reference Beljaars et al. (1989) is gray literature (is this an internal technical report, there is insufficient information), it is not available online, and has limited value here. Provide a better reference.**

We have updated our references regarding the surface fluxes following the reviewer's suggestions.
* * *
**2) Eq. 3: There is again a sign error in this equation. The current formulation wrongly leads to a negative surface-layer bulk Richardson number for stable conditions with $\theta_{v,1} > \theta_{v,1}$ . Such a stable stratification needs to have a positive Ri b number (see also Holtslag and Boville (1993), their Eq. (2.8), for the correct definition). Since Ri b is used in Eq. (5), there is the potential that most of the simulations in this manuscript are wrong. This needs to be clarified.**

The reviewer is right. we have updated the formulation of the equation in the manuscript. This now also corresponds to how it was already implemented in our model code.

> **3) Page 4, lines 4 & 5, and Eq, (3): The definition of $\theta_{v,ref}$ is vague. What do you mean by 'reference value'? Provide the exact definition. Obviously, this reference value of $\theta_{v,ref}$ inthe surface layer must be different than the $\theta_{v,ref}$ values used later in the equation for the planetary boundary layer (Eq. (11). However, the same symbol is used, and no further explanations are offered. Correct this.**

$\theta_{v,ref}$ relates the virtual potential temperature ($\theta_v$) to the buoyancy ($b$), according to the Boussinesq approximation,

$$b = \frac{g}{\theta_{v,ref}} \left( \theta_v - \theta_{v,ref} \right).$$

Meaning that for a dry boundary layer, $\theta_{v,ref}^{-1}$ is supposed to be a sufficiently accurate approximation for the thermal expansion coefficient of the air (Bousinesq). As the equations suggest, it is indeed taken as a constant in our model for the ABL, and for the two GABLS cases they are part of the case set-up definition. We have updated the manuscript to be more clear and added an extra reference that provided some inspiration for our implementations. As the reviewer suggests, the approximations are indeed not generally valid and would be one (of many) improvements that would need to made to have arrive at a more realistic model. We consider this topic to be adressed sufficiently in the manuscript and the previous discussions.

> **4) Page 4, line 8-9: the capital $Z_{0,M}$ symbol is undefined, needs to be $z0,M$ . Provide the value of the roughness length to make the results reproducible.**

The reviewer is right, it should not have been capitalized. The concept of the roughness length is important for the used closure but it's value is typically associated with specific details of the roughness elements at the surface. Therefore, we cannot provide a single value and is also part of the exact case description of the GABLS1 and GABLS2 cases. If one wishes to reproduce the GABLS cases, it would be required to follow the provided references to the original publications of the GABLS intercomparisons. Also, an unambigious description of our methods and implementation are documented and freely available online and may be found by following the links in Table 1 or the section labeled 'code and data availability' which also includes a tutorial on how to install Basilisk and run the code. As such, we are confident that all our results are reproduceable down the binary-representation precision of a computer.

> **5) Eq. (9): Incorrect definition of the vertical wind shear magnitude. It needs to read,**
>
> $$S_{new} = \sqrt{\left( \frac{\partial u}{\partial z} \right)^2 + \left( \frac{\partial v}{\partial z} \right)^2}$$
>
> **instead of the currently used definition**
>
> $$S_{old} = \| \frac{\partial U}{\partial z} \|$$
>
> **If the incorrect formulation has been used in the computations, they will need to be repeated.**

The reviewer is right and we have updated the manuscript accordingly. Note that we had already implemented the suggested formulation in our computations, also taking into account so-called 'directional shear'.

> **6) Eqs. 14, 15, 16: You converted the former vector equation to a scalar equation (as requested) but left the scalar product operator in the formulation. This is mathematically incorrect for the scalar formulation. The dot product needs to be removed. At which time level is the forcing 'r' evaluated? Explain (page 6, line 24) that 'n' denotes the time level.**

The reviewer is right we have updated the manuscript. Which now states that r is integrated in the forward direction, which is indeed an important numerical detail and this warrants a slight textual update as well.

> **7) section 3.1: The description of the Ekman spiral test is insufficient and needs thorough revisions. It is furthermore unclear how it is correctly implemented. I disagree with the author's reply to my first review that it is not necessary to know the values of the parameters. Without the given values of U geo , f (and thereby the latitude angle $\varphi$), $\Omega, \nu, \rho$ the test case is irreproducible. These values need to be provided.**

As mentioned in the text this case bears no resemblance to the planetary ABL, therefore we no not see why the lattitude angle on a rotating sphere would need to be introduced. Furthermore, choosing units and corresponding numerical values for $f, \nu, \rho$ and $U_{geo}$ becomes highly arbitrary. Remarkably (and fortunately) it appears directly from the (steady) analytical Ekman solution (when written as $u_i(\gamma z)/U_{geo} = f_i(\gamma z)$, with index $i$ a dummy for each direction $x, y$) is independent from the value of a single dimensionless group defined as e.g. $\Pi = \frac{U_{geo}}{\gamma \nu} = \frac{2U_{geo}\gamma}{f}$. This makes the results particularly reproducable as they are then universal. This in turn means that *any* combination of values for the variables yields the same results when they are properly scaled. We feared that we would risk conveying a different message when listing numerical values for the chosen parameters. Unfortunately, despite our efforts towards this goal, we did not present such properly scaled results as pointed out by the reviewer in the the next points. Based on the reviewers comments we now also realize that the timestepping parameter and the finite machine precision, which appears to influence the results (see discussion below and Fig 1. of this response), does indeed warrant the introduction of the numerical values for reproducability. Therefore we state that in our set-up we have used normalized values for $U_{geo}, f$ and $\gamma$, meaning that $\nu = 1/2$. Note however that the present results are (now) presented in their proper dimensionless form, and that any unit choice for length, time or velocity will yield equivalent results. Noting that the reproducability of our results is further garanteed by the fact that our exact methods are documented, shared, mirrored and are freely available (GPL v3 license), see text.

> **In addition, it needs to be clarified that (a)$\nu$ is constant (hidden information via the words 'without any closures', is this correct?) and serves the role of K($=\nu$) in Eqs. 14-16**

The reviewer is correct, the revised text is now more explicit on K and $\nu$, and that the test case concerns the *laminar* Ekman spiral.

> **(b) Eqs. (1)-(11) are irrelevant for the discussion**

Yes, using these closures is not convinient when the goal is to test consistency of the numerical methods. We would argue that this is sufficiently made clear and motivated in the text that the equations parameterized turublence in the ABL and that the Ekman spiral case does not concern the ABL.

**(c) the exact definition for the forcing terms r needs to be provided for u and v. In order to arrive at the analytical solutions (17) and (18) of the Ekman spiral, it must be assumed that the motion vanishes at z = 0 and tends to the zonal geostrophic value v = $U_{geo}$ in the free atmosphere. In addition, the Ekman solution (17) and (18) is based on the equation set**

$$0 = K\frac{\partial^2 u}{\partial z^2} + fv - \frac{1}{\rho}\frac{\partial P}{\partial x}$$

$$0 = K\frac{\partial^2 v}{\partial z^2} - fu - \frac{1}{\rho}\frac{\partial P}{\partial y}$$

**When comparing this formulation to Eq. (14) identify exactly how the forcing term r represents the forcing from the Coriolis and pressure gradient term in the u and v equations (provide the equations for r u and r v ). Note that your definition of**

$$\frac{\mathrm{d}P}{\mathrm{d}y} = U_{geo}f\rho$$

**seems to have a sign error and might need to read**

$$\frac{\mathrm{d}P}{\mathrm{d}y} = -U_{geo}f\rho$$

**if you imply geostrophic balance.**

The reviewer is right, and we have revised the manuscript accordingly.

**The other problem with section 3.1 is the incorrect definition of $\gamma$ (line 20) which needs to be**

$$\gamma = \sqrt{\frac{f}{2K}} = \sqrt{\frac{2\Omega\sin(\phi)}{2K}} = \sqrt{\frac{\Omega\sin(\phi)}{K}}$$

**Only in the very special case of $\phi = \pi/2$ (North Pole) is this equation identical to the definition of $\gamma$ in the manuscript. However, this is not specified, and $\gamma$ might be used in an incorrect way**

The reviewer is right. We had stated that the equations of motion are evaluated in a rotating frame of reference with angular velocity $\Omega$. We now realize this is confusing and unclear for two reasons. 1) $\Omega$ only concerned the rotation of our model around it's axis (i.e. the vertical direction), not around the axis that connects the earth's poles. 2) We had not realized that the symbol $\Omega$ is typically associated with the earth's rotation speed. As such the manuscript is revised. The revised manuscript now only uses the Coriolis parameter $f$, and is explicit on how it enters the equations. (see prev. points)

**In addition, the authors call the quantity $\gamma$ 'Ekman depth'. Since the physical units of $\gamma$ are m -1 this is inadequate (it is an inverse). The $\gamma$ definition is then wrongly used in the definitions of z top (line 22), t end and dt. The physical units do not work out. Divisions by $\gamma$ are needed instead of multiplications. The wrong use of $\gamma$ also affects Figs. (1) and (2). The x-axis label $\Delta/\gamma$ must have units of m 2 in the current version (not dimensionless). Figs. 1 and 2 furthermore suggest that U geo = 1 m/s was selected in practice. This is the necessary value to represent the upper error limit of 0.25 m/s along the scaled $\xi/U$ geo y-axis in Figs. 1 and 2. Is this assumption correct? All these aspects need to be clarified.**

The reviewer is right, we have been inconsistent with the usage of $\gamma$ and somehow attributed it with the wrong units, alternating between units of length and inverse length, depending on it's usage. This is addressed

[Figure]

Figure 1: Vertical profiles of the error in the numerically obtained solution ($\epsilon_u$) for the Ekman spiral case at $t = t_{end} = 10/f$ for the run using 128 grid cells. The left hand side (lhs) plot shows the results for runs with three different values for $U_{geo}$ (see legend). Notice the log scale for $\epsilon_u$. The rhs figure plots the same data scaled with the value for $U_{geo}$ to reveal the velocity-scale invariance of our results with respect to it's numerical value. Note that the errors saturate due to the finite precision of the binary representation of numbers.

in the revised manuscript and is now used as an inverse length scale only.

> Figs. 1 and 2 furthermore suggest that U geo = 1 m/s was selected in practice. This is the necessary value to represent the upper error limit of 0.25 m/s along the scaled $\xi$/U geo y-axis in Figs. 1 and 2. Is this assumption correct? All these aspects need to be clarified.

A numerical value of 1 for $U_{geo}$ was indeed used (normalized) and we wonder how the reviewer was able to deduce this from the results? The units for velocity (m/s) are arbitrary as nowhere in the text regarding the case set-up, units for length and time are given. Based on the reviewers comments we have added the used values for $U_{geo}$, $f$ and $\gamma$ in the revised manuscript. However, we remain certain that the value of $U_{geo} = 1$ is not implied by the results of Fig. 1 and 2. To demonstrate the scale invariance of our results we show the correspondence between the results that would be obtained if one would choose a numerical value for $U_{geo} = 5$ or $U_{geo} = 0.001$. The attached figure shows the diagnosed error $\epsilon_u$ for three runs; using $U_{geo} = \{1, 5, 0.001\}$. The domain is discretized using 128 equidistant cells (note that $\gamma$ and $f$ do not change between the runs). Figure 1 of this response shows vertical profiles of the error and reveals that the values of $\epsilon_u$ are higher for the case with the increased wind. However, when we scale the found errors of both runs with the corresponding value for $U_{geo}$, their values are the same (within machine precision)! This means that our results are representative for an infinite set of runs where $U_{geo}$ is varied. For this non-realistic scenario, we prefer the non-dimensionalized presentation of our results as these values can be easily compared with any simulation result where other values may have been used. We hope this sufficiently clarifies the chosen formulation.

> 9) Page 8, line 26: how do the 1000 time steps compare to the setting of t end and dt?

We choose $t_{end} = 10/f$ and $dT = 0.01/f$. The manuscript is now more clear now on this.

> **10) Section 3.2: Point out that this is a dry test case. It looks as if the GABLS1 case only forces the zonal momentum (line 19). Also add the information about the constant Coriolis parameter f and the density $\rho$. Does the density vary with height and if yes, how? As in section 3.1, provide the exact forcing functions r u , r v and r $\theta$ . It seems clear how Eqs. (6)-(11) connect to Eqs. (14)-(16) (via the computation of K), but it is unclear how the surface flux equations (1)-(5) enter Eqs. (14)-(16). Provide this information.**

The reviewer is right and we now also discuss how the surface fluxes enter the computations and how $r$ is formulated in Sect. 2. As for the case descriptions, we have chosen to only provide a brief description of the case and refer to the literature for the details. The density of air does not explicitly enter the equations that are solved in our SCM.

> **Cuxart et al. (2006) presented their results after 9 hours (averaged over the 9 th hour). You average the results over the 8 th hour and compare to Cuxart et al. (2006). What is the reason for the discrepancy? Are the results converged enough to a steady-state solution that the 8 th and 9 th hour time frames become comparable? Provide an explanation.**

The reviewer is right about this flaw, there is no good reason that we presented the 8-th hour average and compare against the 9-th hour average of the LES results of Beare et al. (2006). The GABLS1 case is therefore re-run, for both the equidistant and adaptive grid approaches until $t_{end} = 9h$ to calculate the 9-th hour average. Noting that the resulting profiles for the wind did not change significantly. However, due to the constant cooling rate, the temperature at the surface did change by another 0.25 Kelvin and the near surface results do now *correspond better to the LES results* ... Furthermore, figure 2 now covers the evolution of the grid structure over 9 hours instead of the previous 8 hours and the numbers in table 1 have also been updated. The new results do not warrant any changes to the analysis in the text. We also wanted to visually inspect the steadiness of our solution (and grid), and therefore we have rendered a movie of the evolution of our solution (and grid). The resulting movie is considered nice enough to share and it is available via the link; `https://vimeo.com/284590243`. The soundtrack is composed by Wagner and is added for a dramatic cinematic effect only ;).

> **11) Page 10, line 25 and Figs. 6a,b: which time snapshot is shown? Add this information to the text and the figure caption. The domain is 4000 m high, but only 1300 m are shown in Fig. 6? Why? How do the solutions compare in the upper domain?**

We have added the information regarding the time. For the comparison we follow the analysis of the GABLS2-intercomparison participants as in Svensonn et al. (2011). We can only *guess* their motivations; Maybe they were (like us) most interested in the representation of the atmospheric boundary layer and not the free troposphere aloft. Or the variations between the models were small as the SCM has not much dynamics there. Anyhow, as mentioned in the text, we obtained their data from their plots that only covers the ABL, so we cannot compare at higher altitudes. This latter argument does not apply to figure 7, so based on the reviewers comments, we updated *that* caption to state that the profile of $U$ is constant with height for $z > 1200m$.

Correction of typos and style:
Page 1, line 15: '... an SCM ...'
Page 2, line 27: ' ... built-in ...'
Page 4, line 14: '.. description ...'
Page 5, line 24 and page 7, line 25: '... its ....'
Page 6, line 17: '... spent ...'
Page 7, line 10: should read ' ...  clean setup quantifies numerical errors explicitly and tests the ...'
Page 8, line 2: Bring footnote into the main text
Page 8, line 3: '...shows the results of the errors at all levels and ...'
Page 8, line 15: '... though...'
Page 8, line 28: '... arise in the solution ...'
Page 9, line 1: '...and the computational performance ...'
Page 10, line 1: '... parameterize ...'
Page 10, line 6: ... on the order of ...'
Page 11, line 2: 'Fig. 5' needs to read Fig. 8
Page 11, line 8: '... presented a one-dimensional ...'
Caption Fig. 1 and 2: Add the information that the errors are shown at t end (the end of the simulation). Also add: the inset shows the errors for all time steps.
Fig. 3: symbol 'L' is undefined
Caption, Fig. 4: '... eighth hour ...'. Do u and v stay constant above 275 m?

The revised manuscript has taken all the suggestions into account. Except for the suggestion to add that the inserts of Fig. 1 and 2 show the errors for all time steps. The insert show the same data as is plotted in the main plot, but then for a single run. This was done to prevent that data being obscured by subsequently plotted dots of the other runs, and thereby reveal the full spread of this typical result for a single run.

At the risk of repeating ourselves, we again like to thank the reviewer for bringing forward the discussed issues. We feel He or She has greatly helped to improve the quality of the manuscript.

[revised manuscript text omitted]

---

## Author Response (AR3)

**Response to the Reviewer**

Antoon van Hooft, Stéphane Popinet, and Bas van de Wiel

October 2018

Without any exeption, the authors agree with the points brought forward by the reviewer and a point-by-point reponse is given below. Furthermore, an overview of the changes made to the manuscript are presented via the highlighted markup.

**1 Response to Reviewer #3**

The authors thank the reviewer for his/her continued efforts to help improve the manuscript.

> **The newly revised manuscript (September 2018) has greatly improved, answered almost all of the questions from the 2nd review, and cleaned up most of the erroneous equations. However, two sign errors remain in the current version of the manuscript. As before, the authors need to check whether these sign errors were present in the calculations, or whether these are typos in the manuscript. In case the errors are in the code, the results need to be reproduced. I also have a few more clarifying questions, and suggest a few corrections of the text.**

We are happy to learn that the manuscript appears to converge from containing $\mathcal{O}(10)$ sign errors in the 1st revision, via $\mathcal{O}(1)$ to hopefully $\mathcal{O}(0)$ sign errors in the most recent iteration. Again, the sign errors are of typographical nature only and the calculations were performed using the correct formulations.

> **1) Page 4, lines 4 & 5, and Eq. (3) and page 5, Eq. (12): As I pointed out in my 2nd review the definition of $\theta_{v,ref}$ is vague. My earlier review asked what you meant by 'reference value'? The authors provided a satisfactory answer in their reply to my review comment, but none of this information made it into the manuscript. For example, the reply stated that a constant $\theta_{v,ref}$ value (however, it is not provided) is used in the ABL case, and that the GABLS tests define this reference profile. Therefore, the manuscript still lacks clarity. It makes it impossible to reproduce the ABL results (provide $\theta_{v,ref}$), and the authors should point to the GABLS tests for their specific $\theta_{v,ref}$ definitions. Please add the information from the reply to the manuscript.**

The revised manuscript now explains the physical meaning of $\theta_{v,ref}$ and lists their values at the (short) descriptions of the cases. Noting that the sake of focussing the present manuscript, we still refer the interested reader to the original GABLS papers for the complete case descriptions.

> **2) Page 5, Eq. (13) and line 17: Do you refer to the same spatially- and time-dependent $K$ coefficient that is defined in Eq. (8)?**

Yes, this is now stated explicitly in the revised manuscript, and this naturally arises from Eq. 7 that expresses the flux as a function of $K$.

> **3) Page 5, Eq. (15): The K diffusion coefficient as defined in Eq. (8) has a time dependency. Indicate in Eq. (15) whether K gets evaluated at time level n or the future time level n+1.**

The revised manuscript is now clear that indeed $K^n$ is used, making the time integrator formally a mixed implicit-explicit method. Consistency of the method relies on the smooth evolution of K and that is in-line with the assumption that the scalar fields for $u, v$ and $\theta_v$ evolve sufficiently smooth such that a numerical

time-integrator may be used to solve the problem. Which is a very generic assumption

> **4) Page 6, line 5: the expression for the 'pressure gradient force vector' must be $-\frac{1}{\rho}\nabla p$ , not just $-\nabla p$.**

The reviewer is right and hence the manuscript is revised accordingly.

> **5) Page 6: I had already pointed out the sign error in the geostrophic relationship in my 2nd review, but these errors are still present. The right hand side of Eq. (17) must read $-\frac{1}{\rho}\nabla p - f\mathbf{k} \times \mathbf{u}$ and Eq. (18) must read $U_{geo} = \frac{1}{\rho f}\mathbf{k} \times \nabla p$ . If the sign errors are present in the code, the results need to be reproduced.**

The reviewer is right and the equations have been updated with the correct signs. We have checked to code and this mistake was was not present in the calculations. Noting that during the development phase, sign errors were also made. Fortunately, errors of such sort get readily identified by running tests and then obtaining unphysical numerical solutions.

> **6) Page 6, line 9: It is incorrect to define the symbol $\times$ as the 'vector outer product operator' (which leads to a matrix). The symbol is the 'cross product' (leading to a vector).**

The reviewer is right and we have updated the manuscript.

> **7) Page 7, end of line 10: typo, should read '... of a choice ...'**

The manuscript is revised accordingly.

[revised manuscript text omitted]

---

## Author Response (AR4)

Dear James R. Maddison,

Thank you for your continued efforts,

Topical Editor Decision: Please could you add an "Author Contribution" section to this paper, as described in the manuscript preparation guidelines.

Typos: "are rule" (abstract), "whos" (below equation 3), "it's" in place of "its"

We have updated the manuscript following all your suggestions. While writing the "author contribution" section, we realized we had forgotten to acknowledge the people who contributed to the GNU project, this is now also fixed.

See the highlighted changes to the manuscript below.

Antoon van Hooft, Stephane Popinet and Bas an de Wiel

[revised manuscript text omitted]

---

## Author Response (AR5)

Dear Editor,

We have removed the last, gramatically incorrect, sentence from the author contribution section.

On behalf of all Authors; Thank you for your efforts towards our manuscript.

Antoon van Hooft

changes:

....

*Author contributions.*
All Authors contributed to the content of this manuscript and it should be
viewed as a fruit of the many discussions we have had. Furthermore, SP wrote the
Basilisk code, he also designed and implemented the grid adaptation algorithm.
The numerical experiments were set up and performed by AvH. The writing was led
by AvH and organized in an iterative procedure with BvdW. ~~Noting that the
authors have no illusion they (can) do justice to the attribution of all the
ideas and input.~~

....